# Structure-activity relationships of mitochondria-targeted tetrapeptide pharmacological compounds

Wayne Mitchell[1†], Jeffrey D Tamucci[1], Emery L Ng[1], Shaoyi Liu[2], Alexander V Birk[3], Hazel H Szeto[2], Eric R May[1], Andrei T Alexandrescu[1], Nathan N Alder[1]*

[1]Department of Molecular and Cell Biology, University of Connecticut, Storrs, CT, United States; [2]Social Profit Network, Menlo Park, CA, United States; [3]Department of Biology, York College of CUNY, New York, NY, United States

*For correspondence:
nathan.alder@uconn.edu

Present address: †Department of Medicine, Brigham and Women's Hospital, Harvard Medical School, Boston, MA, United States

**Abstract** Mitochondria play a central role in metabolic homeostasis, and dysfunction of this organelle underpins the etiology of many heritable and aging-related diseases. Tetrapeptides with alternating cationic and aromatic residues such as SS-31 (elamipretide) show promise as therapeutic compounds for mitochondrial disorders. In this study, we conducted a quantitative structure-activity analysis of three alternative tetrapeptide analogs, benchmarked against SS-31, that differ with respect to aromatic side chain composition and sequence register. We present the first structural models for this class of compounds, obtained with Nuclear Magnetic Resonance (NMR) and molecular dynamics approaches, showing that all analogs except for SS-31 form compact reverse turn conformations in the membrane-bound state. All peptide analogs bound cardiolipin-containing membranes, yet they had significant differences in equilibrium binding behavior and membrane interactions. Notably, analogs had markedly different effects on membrane surface charge, supporting a mechanism in which modulation of membrane electrostatics is a key feature of their mechanism of action. The peptides had no strict requirement for side chain composition or sequence register to permeate cells and target mitochondria in mammalian cell culture assays. All four peptides were pharmacologically active in serum withdrawal cell stress models yet showed significant differences in their abilities to restore mitochondrial membrane potential, preserve ATP content, and promote cell survival. Within our peptide set, the analog containing tryptophan side chains, SPN10, had the strongest impact on most membrane properties and showed greatest efficacy in cell culture studies. Taken together, these results show that side chain composition and register influence the activity of these mitochondria-targeted peptides, helping provide a framework for the rational design of next-generation therapeutics with enhanced potency.

## Editor's evaluation

A number of S-S (Szeto-Schiller) tetrapeptides are known to be targeted to mitochondria. This study shows for the first time a structure-activity relationship for these peptides in reversing mitochondrial membrane potential and ATP loss in stressed cell models. In particular, peptides containing indole residue in the side chain, such as Tryptophan are more active. These results therefore provide valuable new insight on the development of new therapeutics for the treatment of mitochondrial dysfunctional diseases.

## Introduction

As regulators of energy metabolism, mitochondria house the oxidative phosphorylation (OXPHOS) complexes that produce >90% of cellular ATP. Mitochondria also coordinate key cellular processes including lipid biosynthesis, ion homeostasis, and cell death. Consequently, mitochondrial dysfunction, particularly in tissues with high energy demand, is central to the etiology of many complex pathologies including cancer, cardiopathy, neurodegeneration, aging-related ailments, and heritable (primary) mitochondrial disease. Despite this, there are currently no FDA-approved therapeutics for the treatment of mitochondrial diseases.

Mitochondria-targeted cationic-aromatic tetrapeptides are among the most promising pharmacological interventions under development for the treatment of mitochondrial dysfunction. Also termed Szeto-Schiller (SS) peptides, these first-in-class compounds are synthetic C-terminally amidated tetrapeptides with a motif of alternating cationic and aromatic residues that is thought to be important for their ability to traverse membranes in a variety of cell types and to concentrate in mitochondria (*Zhao et al., 2005*; *Zhao et al., 2003*; *Zhao et al., 2004*). Many in vitro, preclinical, and clinical studies, primarily with the lead compound SS-31 (elamipretide), support the therapeutic efficacy of these peptides. Studies with isolated mitochondria and cell cultures show that SS-31 improves electron transfer efficiency and increases ATP production while reducing electron leak and reactive oxygen species production (*Zhao et al., 2004*; *Birk et al., 2014*; *Birk et al., 2013*; *Siegel et al., 2013*; *Anderson et al., 2009*). Animal studies have demonstrated the ability of SS-31 to maintain cellular bioenergetics under stress conditions such as ischemia, hypoxia, and aging-related dysfunction (*Allen et al., 2020*; *Campbell et al., 2019*; *Szeto, 2018*; *Zhang et al., 2020*). The clinical efficacy of SS-31 has been demonstrated for primary mitochondrial disorders (*Zhao et al., 2017*; *Reid Thompson et al., 2021*) and for age-related chronic diseases associated with mitochondrial dysfunction (*Roshanravan et al., 2021*).

Progress toward elucidating the molecular mechanism of action (MoA) of these peptides has come on several fronts. Early studies suggested that SS peptides target the lipid bilayers of mitochondrial membranes through interactions with cardiolipin (CL) (*Birk et al., 2014*; *Birk et al., 2013*; *Birk et al., 2015*), the anionic phospholipid that is enriched in the inner mitochondrial membrane (IMM) and required for proper membrane morphology and function of membrane-bound complexes (*Paradies et al., 2019*). This bilayer-mediated mechanism is supported by work with model systems in which peptide inhibited peroxidase activity of cytochrome *c* (*Birk et al., 2014*) and improved cristae ultrastructure (*Szeto and Liu, 2018*). Recent work from our group quantitatively evaluated SS-31 interactions with CL-containing membranes, showing that the peptide affected lamellar bilayer properties (e.g. lipid lateral diffusion and packing interactions), with the most notable effect being on membrane electrostatics based on down-regulation of the surface potential ($\psi_s$) that originates from the negatively-charged membrane interface (*Mitchell et al., 2020*).

Other recent work has focused on the interactions of SS-31 with mitochondrial proteins. Based on a crosslinking/mass-spectrometry approach with a biotinylated SS-31 analog, the SS-31 interactome was shown to include a subset of membrane complexes primarily involved in ATP-generating processes (*Chavez et al., 2020*). Moreover, in aged cardiomyocytes, SS-31 was shown to reduce proton leak mediated by the adenosine nucleotide transporter (ANT1) and stabilize the ATP synthasome (*Zhang et al., 2020*). Notably, almost all of these SS-31-interactive proteins are known to themselves bind CL. Therefore, mitochondrial lipid composition plays a key role in modulating the molecular interactions of SS-31.

Despite these insights, the lack of information relating the structure and function of these mitochondria-targeted tetrapeptides presents a barrier to a full understanding of their MoA. An effective strategy to address this mechanistic knowledge gap is to test the effects of expanding the sequence space of mitochondria-targeted peptides using structure-activity analyses. In this study, we evaluated three sequence-variant peptides that differed with respect to aromatic side chain content and cationic/aromatic register, compared with the SS-31 benchmark. Our results provide a direct assessment of the tetrapeptides with respect to their membrane-bound conformations, effects on membrane properties, and relative efficacies in preserving cellular viability under stress. This work reveals that side chain composition has a profound effect on the structure and activity of these mitochondria-targeted peptides. Notably, the analog containing tryptophan side chains had the greatest potency in cell stress models, which we can correlate with many of its molecular-level

interactions and effects on reductionist membrane systems. This work establishes potential directions for the rational design of next-generation tetrapeptide analogs with enhanced efficacy as mitochondrial therapeutics.

## Results

### Tetrapeptide analog set: design and rationale

In this study, we compared a test set of four tetrapeptides with different sequences (*Figure 1A*). An alphabet of two basic residues (Arg and Lys) and three aromatic residues (Phe, Tyr and Trp) gives $3^2 \times 2^2 \times 2 = 72$ possible sequence permutations with an alternating aromatic (φ)/basic (B) sequence periodicity (B-φ-B-φ or φ-B-φ-B). However, the number of sequences becomes much larger if D-amino acids (which can extend the medicinal lifetimes of peptides) and/or unnatural amino acids (which increase functional versatility) are included. A large library of peptides precludes detailed structural and functional studies, so we focused on a limited test set of peptides to investigate two fundamental properties: (i) the side chain register (B-φ-B-φ vs. φ-B-φ-B) and (ii) the types of aromatic side chains. The cationic/aromatic register has potential structural ramifications, e.g., in determining the polar interactions between peptide basic groups and CL. Aromatic amino acid type can modulate hydrophobicity, aromaticity, polarity, and hydrogen bonding capacities, which in turn can affect both peptide structure and peptide-membrane interactions (*McDonald and Fleming, 2016*).

Our test set (*Figure 1A*) included two analogs with B-φ-B-φ register, SS-31 (our benchmark), and SPN4. These two analogs differ only with respect to the second-position aromatic residue. SS-31 contains the unnatural amino acid 2,6-dimethyl tyrosine (Dmt), known to be important for free radical scavenging by this peptide (*Zhao et al., 2005*; *Zhao et al., 2004*; *Jiang et al., 2020*). By contrast, SPN4 replaces Dmt with Tyr. Proteinogenic Tyr also has the phenolic OH group that can scavenge radicals and mediate H-bond interactions but allows us to evaluate the effect of the two tyrosine methyl groups on peptide structure and function. Additionally, our test set included two analogs with φ-B-φ-B register, SS-20, and SPN10. With its Phe/Phe aromatic composition, SS-20 does not possess the free radical scavenging properties of SS-31; however, SS-20 is known to target mitochondria (*Liu et al., 2018*; *Alta et al., 2017*) and has also demonstrated efficacy with many mitochondrial disease models (*Cho et al., 2007*; *Szeto et al., 2015*; *Yang et al., 2009*; *Sun et al., 2020*). This confirms that scavenging activity is not an essential feature of the MoA of this class of compounds. Finally, SPN10 is unique in our test set in that it contains only L-enantiomer side chains, and it has two Trp residues that combine a bulky bicyclic indole group with a pyrrole-like NH that can mediate H-bond interactions.

### The free peptides are extended but have some residual structure due to aromatic interactions

As a first step toward comparing the four peptides, we determined their NMR structures in solution. Small water-soluble and membrane-active peptides are typically disordered in solution and adopt their bioactive conformations only upon binding membranes (*Avci et al., 2018*; *Kabelka and Vácha, 2021*). However, even very short peptides can have preferred conformations in aqueous solution (*Jas et al., 2019*), particularly if they are enriched in aromatic residues. Structural analysis of these peptides in solution can shed light on their properties in the extracellular milieu, the cytosol, and aqueous mitochondrial subcompartments.

The NMR structures of peptides in solution are relatively disordered extended conformations (*Figure 1B*). NMR assignments (see *Supplementary file 1*) show few, if any, nuclear Overhauser effects (NOEs) (shown in *Figure 1—figure supplement 1*, panel A), which is consistent with their molecular masses of ~600 Da, as this is near the zero-crossing point for the NOE (*Williamson, 2009*). Furthermore, using SS-20 as a representative peptide, we used pulse field gradient (PFG) NMR diffusion experiments (*Whitehead et al., 2022*) to confirm that the free peptides are in a monomeric state under the conditions used for NMR analysis (see *Figure 1—figure supplement 2*). To characterize distance contacts in solution, we therefore collected rotating frame nuclear Overhauser effect spectroscopy (ROESY) data (*Williamson, 2009*), as the signs of crosspeaks in this experiment are invariant to molecular size (see *Figure 1—figure supplement 1*, panel B). We obtained roughly 30–50 distance constraints per peptide for NMR structure calculations (see *Supplementary file 2*). Notably, although the free peptide NMR structures are relatively disordered (*Figure 1B*), we did observe some residual

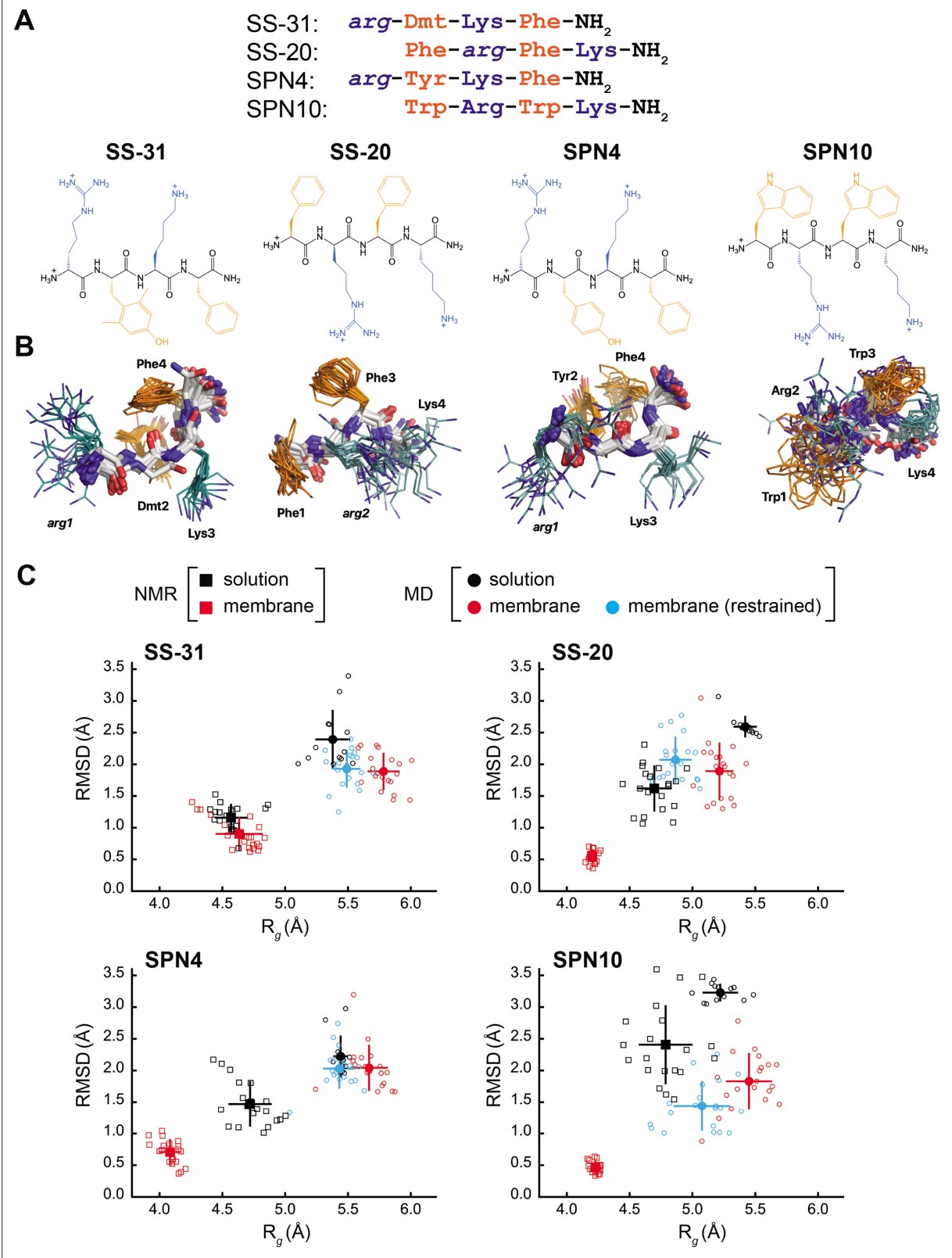

**Figure 1.** Peptide structures and membrane-driven peptide folding. (**A**) Peptide sequences and chemical structures. Upper series: peptide sequences aligned with respect to basic and aromatic side chains; lower series: chemical structures of peptides. Basic and aromatic groups are shown in teal and orange, respectively. Three letter amino acid codes include those for L-amino acids (first letter capitalized, normal font) and D-amino acids (all lower case, italicized). (**B**) NMR structures of peptides in aqueous solution. The top 20 lowest-energy NMR conformers calculated from rotating frame nuclear

*Figure 1 continued on next page*

*Figure 1 continued*

Overhauser effect spectroscopy (ROESY) analysis. The main chain atoms are colored by type (carbon, white; oxygen, red; nitrogen, blue); basic and aromatic side chains are colored in teal and orange, respectively. (**C**) Comparison of peptide conformational variance in NMR and molecular dynamics (MD) simulation assemblies. Radius of gyration ($R_g$) and root mean square deviation (RMSD) to the average peptide structure calculated for NMR structures in solution (black squares) and in the presence of a membrane (red squares), and for MD simulations of peptides in solution (black circles), in the presence of a lipid bilayer conducted without (red circles), and with (cyan circles) nuclear Overhauser effect (NOE) restraints. Measurements of individual peptides (n=13 for MD solution datasets; n=20 for all other datasets) are shown as small symbols, with ensemble averages and error bars (SD) shown as larger symbols. Corresponding values for the $R_g$ and RMSD measurements are in *Supplementary file 4*.

The online version of this article includes the following source data and figure supplement(s) for figure 1:

**Source data 1.** NMR and MD radius of gyration ($R_g$) and root mean square deviation (RMSD) values.

**Figure supplement 1.** NMR cross-relaxation experiments for the SS-31 peptide.

**Figure supplement 2.** NMR pulse field gradient (PFG)-diffusion data showing the SS-20 peptide is a monomer.

**Figure supplement 3.** Root mean square deviation (RMSD) of peptides in solution from MD trajectories.

structure influenced by the side chain register. Specifically, tetrapeptides with a B-φ-B-φ motif (SS-31 and SPN4) had lower root mean square deviation (RMSD) values (better structural precision) and three non-sequential NOEs, whereas those with a φ-B-φ-B motif (SS-20 and SPN10) lacked non-sequential NOEs (*Supplementary file 2*). Importantly, the non-sequential NOEs in SS-31 and SPN4 occurred between the two aromatic residues, consistent with previous observations that residual structure is more common in short peptides containing aromatic side chains (*Neri et al., 1992*; *Smith et al., 1994*). Additional interactions that appear to stabilize the tetrapeptide structures in solution are cation-π interactions between neighboring basic and aromatic amino acids. These are supported by significant upfield ring current shifts for some of the basic residue side chain nuclei (*Supplementary file 1*), for all peptides except SS-31.

As a second approach to assessing peptide solution structures, we performed all-atom MD simulations (200 ns each) of the peptides in an aqueous environment. To initiate each simulation, the solution structure of each peptide closest to the average structure of the NMR ensemble was used. To directly compare the ensemble of structures from our NMR- and MD-based approaches, we calculated the RMSD to the ensemble average, as well as the radius of gyration ($R_g$) of each peptide. The MD-derived ensembles had greater structural variability (higher RMSD) and were less compact (higher $R_g$) than their cognate NMR-derived structures (*Figure 1C*, compare black squares and circles). However, consistent with our NMR results, the RMSD values calculated by MD simulations were lower for SS-31 and SPN4 than for SS-20 and SPN10 (see *Figure 1—figure supplement 3*). Notably, all of the RMSDs from membrane MD simulations were lower than the limiting RMSDs for tetrapeptides in solution MD simulations, which can deviate up to 5 Å from the initial structures (see Materials and methods). Taken together, the results of our NMR and MD analyses suggest that the free peptides are largely disordered but retain some residual structure due to cation-π and aromatic ring stacking interactions. We next proceeded to empirically evaluate the interaction of our tetrapeptides with biomimetic membranes.

## Tetrapeptide analogs have distinct equilibrium binding behavior to CL-containing membranes

To assess the membrane-binding properties of the peptide analogs, we performed isothermal titration calorimetry (ITC) of peptides titrated with large unilamellar vesicles (LUVs) containing a 80:20 molar ratio of 16:0/18:1 phosphatidylcholine (POPC) and 18:1 CL (tetraoleoyl-CL, TOCL) (*Figure 2*). This approach provides a full thermodynamic characterization of the peptide-membrane interaction. Fits to binding isotherms (*Figure 2A*) provided equilibrium binding parameters (*Figure 2B*) that revealed key similarities and differences among the membrane-interactive properties of our test set. First, all tetrapeptides bound CL-containing membranes with roughly similar binding affinity ($K_D$ 27.5 μM to 39.5 μM; $\Delta G$ –26.2 kJ/mol to –25.9 kJ/mol). These binding affinity values did not differ significantly among the analogs. However, the lipid-to-peptide binding stoichiometry, $n$, did differ among peptides. To calculate this value, we used the 'effective' lipid concentration ([lipid]$^{eff}$) that represents lipids on the outer leaflet of the liposomes to which peptides have access (*Mitchell et al., 2020*). Compared with the benchmark SS-31 (n=5.4; i.e. an average of 5.4 lipids per bound peptide), SPN4

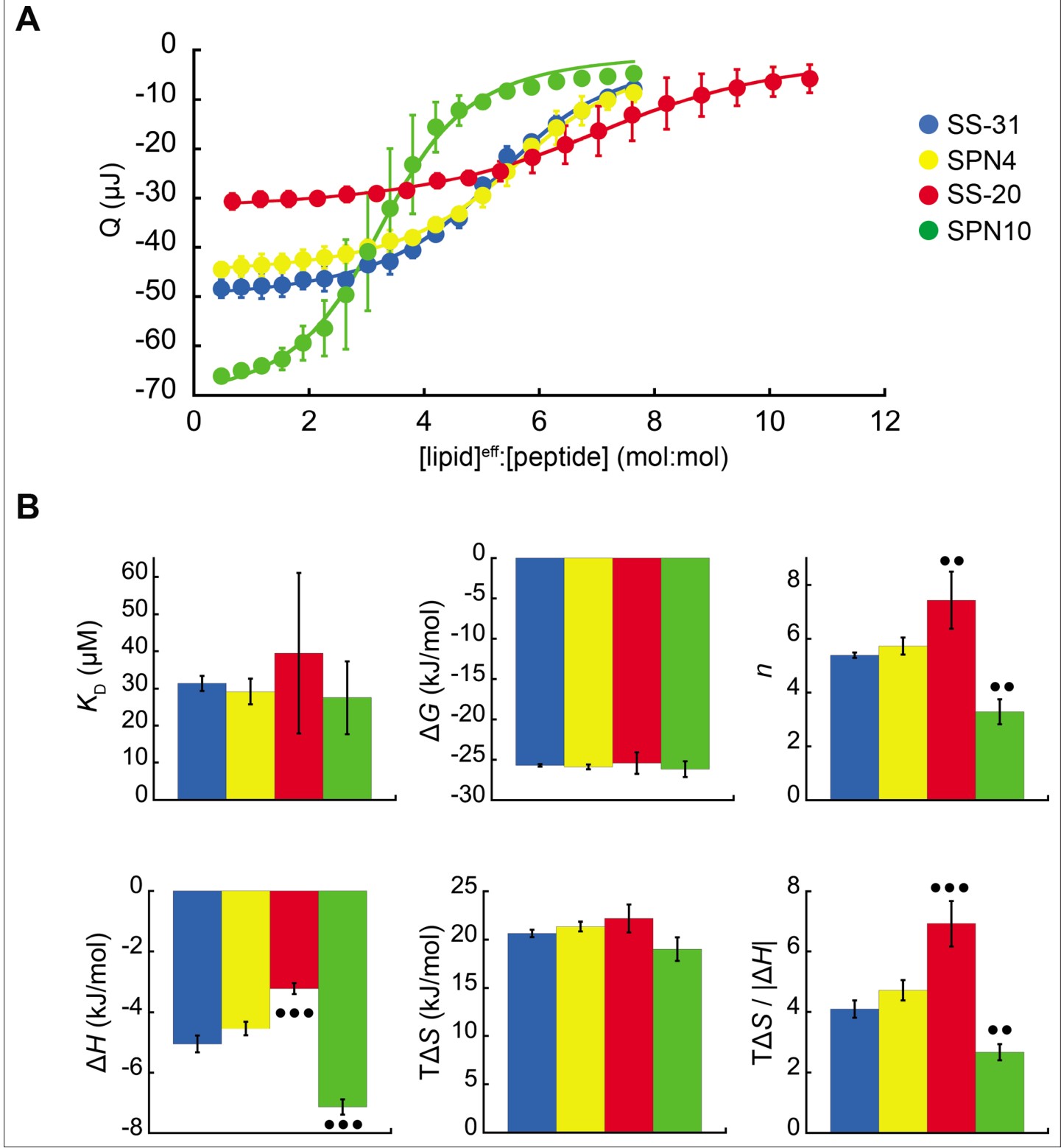

**Figure 2.** Microcalorimetry of peptide binding to large unilamellar vesicles (LUVs). (**A**) Equilibrium binding isotherms. Wiseman plots showing blank-corrected average integrated heats for lipid-into-peptide titrations as a function of [lipid]$^{eff}$:[peptide] molar ratio using LUVs composed of an 80:20 molar ratio of phosphatidylcholine (POPC):tetraoleoyl cardiolipin (TOCL) with peptides color-coded as shown. Points represent means (n≥3 ± SD), and curve fits are to binding models of single, independent sites. (**B**) Comparison of binding parameters. Values of equilibrium binding parameters calculated from

*Figure 2 continued on next page*

*Figure 2 continued*

curve fits are shown for peptides color-coded as in panel A. Statistical comparisons represent one-way ANOVA with Tukey's multiple comparison test (α=0.05), with differences representing a comparison to SS-31 (no symbol, p>0.05; •• p≤0.01; ••• p≤0.001).

The online version of this article includes the following source data and figure supplement(s) for figure 2:

**Source data 1.** Isothermal titration calorimetry data and equilibrium binding parameters.

**Figure supplement 1.** Peptide analog binding footprints and membrane area coverage.

had a similar value (n=5.7), SS-20 had a significantly higher value (n=7.4), and SPN10 had a significantly lower value (n=3.3). These results indicate that SPN10 and SS-20 bind membranes, respectively, at higher and lower surface densities than SS-31.

Membrane interaction of all peptides was enthalpically favorable (ΔH<0) and entropically favorable (TΔS>0), with binding dominated by entropy as observed previously for SS-31 (*Mitchell et al., 2020*). In comparison with the binding enthalpy of SS-31 (ΔH=–5.1 kJ/mol), SPN4 had a similar value (ΔH=–4.5 kJ/mol), whereas the magnitude of binding enthalpy was significantly lower for SS-20 (ΔH=–3.2 kJ/mol) and higher for SPN10 (ΔH=–7.1 kJ/mol). As ΔH is a function of polar contacts made during binding (*Seelig, 2004*), this trend in ΔH is consistent with the number of aromatic side chains containing polar groups (SPN10, two indole NH groups; SPN4/SS-31, one phenol OH group; SS-20, none). Binding entropy, which in this system is largely a function of aromatic side chain partitioning into the acyl chain region (*Seelig, 2004*), did not differ significantly among peptides (TΔS ranged from 19.0 to 22.1 kJ/mol). However, the relative enthalpic and entropic contributions to binding, quantified by the ratio TΔS/|ΔH|, indicate that in comparison with SS-31, membrane binding of SS-20 is more entropy-driven and that of SPN10 is more enthalpy-driven. Having established these differences in membrane-binding behavior among our peptide analogs, we then investigated their structural differences in the membrane-bound state.

## The bicelle-bound peptides adopt distinct reverse turn structures, except for SS-31

To determine the structural features of cationic-aromatic tetrapeptides in a membrane-like environment, we used bicelles as membrane mimetics. We hypothesized that moving from a high to a low dielectric environment would promote a more uniform structure and that the measured differences in the membrane-binding thermodynamics of these peptides could have a structural basis. As previously reported (*Birk et al., 2014*), in the presence of CL-containing bicelles, the NMR signals of SS-31 broaden at low peptide:lipid ratios and then sharpen again at a molar excess of peptide to CL. This indicates that the free and bicelle-bound states of the peptides are in fast exchange on the NMR time-scale and should be amenable to transferred NOE (trNOE) studies (*Post, 2003*). The structures of all four peptides bound to bicelles (*Figure 3*) were calculated based on a large number of negative NOEs, consistent with a high MW peptide-bicelle complex, that are transferred to the free peptide (see *Figure 1—figure supplement 1*, panel C; *Figure 3—figure supplement 1*; and *Supplementary file 3*). The NMR structures are precise because they are each defined by 95–110 trNOEs per tetrapeptide or about 25 structural restraints per residue (*Supplementary file 3*). This is reflected in heavy atom RMSDs values of 0.5–0.9 Å for the bound peptides, which is less than half of those of the free peptides (compare *Figure 1C* and *Supplementary file 4*). NMR structures of all the peptide analogs had lower RMSD and $R_g$ values in the membrane-bound relative to the free state. In other words, they became more structurally constrained and compact upon binding (*Figure 1C*, compare black and red squares). The exception was SS-31, whose RMSD and $R_g$ values did not statistically change upon binding.

Interestingly, we found that in the membrane-bound state, all peptide analogs except for SS-31 formed H-bonded reverse turn structures with basic side chains pointing away from the plane of the backbone ring (*Figure 3A*). This gives the peptides a markedly asymmetric charge distribution (shown in *Figure 3—figure supplement 2*), with the cationic face of the peptides likely poised for binding to the negatively charged lipid phosphates of CL-containing membranes. To form the reverse turn structures, the φ-B-φ-B peptides have CO(1) to NH(4) H-bonds. However, for the B-φ-B-φ peptide SPN4, the H-bond is formed with the capping NH₂ group that essentially acts as the amide proton donor of a non-existent fifth residue in a CO(2) to NH₂(5) pattern (*Figure 3B*). In contrast, the other

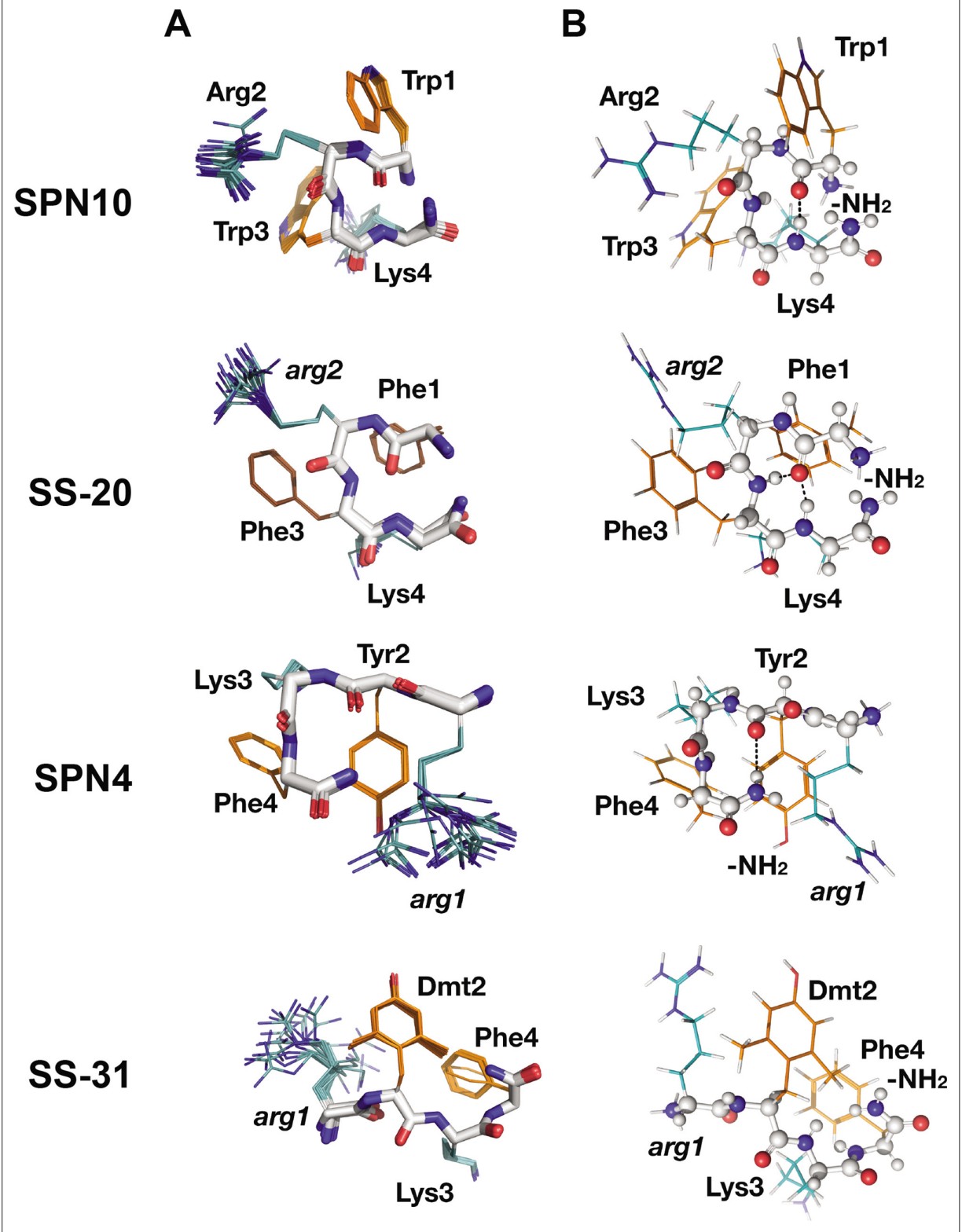

**Figure 3.** Peptide structures in the bicelle-bound state. (**A**) Top 20 lowest energy NMR conformers. Backbone atoms and side chains are color coded as in *Figure 1B*. (**B**) Peptide secondary structure. Conformations closest to NMR ensemble means are shown. Backbones are shown in ball-and-stick representation, H-bonds are shown as dotted lines, and side chains are shown as lines (orange for aromatic and teal for basic). Three of the peptides have main chain H-bonds: SPN10 (CO[1] to NH[4]), SS-20 (CO[1] to NH[4]), and SPN4 (CO[2] to NH₂[5]). SS-31 is extended and has no H-bonds.

The online version of this article includes the following source data and figure supplement(s) for figure 3:

*Figure 3 continued on next page*

*Figure 3 continued*

**Figure supplement 1.** Representative sections of 2D NOESY spectra.

**Figure supplement 2.** Electrostatic surfaces of peptides in the bound state.

**Figure supplement 3.** Assignments of bicelle lipid NMR signals.

**Figure supplement 4.** Transferred nuclear Overhauser effects (trNOEs) between peptides and lipids.

**Figure supplement 5.** Transferred nuclear Overhauser effect (trNOE) buildup curves.

**Figure supplement 5—source data 1.** Transferred nuclear Overhauser effect (trNOE) buildup curve data.

B-φ-B-φ peptide SS-31 adopts an extended conformation due to steric restraints induced by the methyl groups on Dmt2 that preclude the turn conformation from forming. For the peptides that form a reverse turn, intra-peptide cation-π interactions are observed, and the backbone H-bond NH donor in the turn structure is always an aromatic residue. For the φ-B-φ-B peptides SS-20 and SPN10, cation-π interactions form between Arg2, Phe3 and Trp3, Lys4, respectively. Although these cation-π interactions may arise simply from sequence proximity, they could play a critical role in reducing the overall polar character of these peptides, allowing them to reside within the low-dielectric membrane interface. As for the basic residues, Arg is always the more poorly defined side chain in the structures, and Lys is generally precisely defined, suggesting that the latter might be experiencing restricted motion due to partial insertion into the membrane (*Figure 3A*).

We also evaluated the membrane-bound structures of the four tetrapeptides using MD simulations, where multiple peptides were allowed to associate with a bilayer consisting of an 80:20 molar ratio of POPC:TOCL (*Figure 4A*). All peptides rapidly adsorbed to the membrane surface within 500–750 ns and evolved toward a stable, bound configuration over the course of the 2 µs simulations (see *Figure 4—figure supplements 1–5*). As with the solution structures, membrane-bound structures from MD had higher RMSD and $R_g$ values than those from NMR (compare *Figure 1C* and *Supplementary file 4*).

To further compare our membrane-bound MD and NMR structures, we calculated the average fraction of time each peptide in the simulation was within 3 Å heavy atom RMSD to the NMR top structure. Overall, the MD ensembles had a low frequency of sampling states with high similarity to the NMR-derived structures (*Figure 4B* and *Supplementary file 5*). This low degree of structural overlap motivated the extension of each simulation for an additional 1 µs, during which NOE distance restraints between residues *i* to *i+2* and *i* to *i+3* were imposed for each peptide (*Abraham et al., 2015*; *Hess et al., 2008*; *Hess and Scheek, 2003*). Imposing the NMR restraints yielded considerable improvements in structural similarity to the NMR structure for all peptides (*Figure 4B* and *Supplementary file 5*). NOE violations were computed over the MD simulations using time and ensemble averaging and violations occurred less than 20% of the time for the majority (~71%) of the applied restraints (*Figure 4—figure supplement 6*). Thus, all subsequent MD analyses were performed using NOE restrained MD data, unless otherwise indicated.

## Peptide analogs have different membrane insertion profiles and lipid interactions

We next analyzed our MD simulations for the membrane insertion depth of each peptide side chain. Side chain insertion depth ($Z^{pos}$) for each peptide analog was measured by averaging the distance in the Z-direction (bilayer normal) from each side chain's $C^\beta$ atom to the bilayer center of mass (COM) (*Figure 4C* and *Figure 4—figure supplements 1–5*). Generally, the residues of SS-31 tended to bury deepest, followed sequentially by SPN10, SS-20, and SPN4.

In the trNOE experiments, we also observed NOEs between the peptides and the bicelle lipids, for which we used lipid proton assignments from the literature (see *Figure 3—figure supplements 3–4*; *Liebau et al., 2017*). Most of the peptide-lipid NOEs were from aromatic residues to lipid protons close to the headgroup region. This suggests peptides are superficially buried in the interfacial region of lipid bilayers, in agreement with the MD results. Interestingly, the peptide-lipid NOEs for SPN10 were generally weaker than those of the other peptides (*Figure 3—figure supplement 4*). Although we saw few, if any, trNOEs from lipid to the basic groups, this could be explained by the fact that the TOCL $^1$H signals were largely overlapped and dominated by the ~40-fold molar excess of DHPC and POPC (*Figure 3—figure supplement 3*). Additionally, given the 1/r distance dependence of

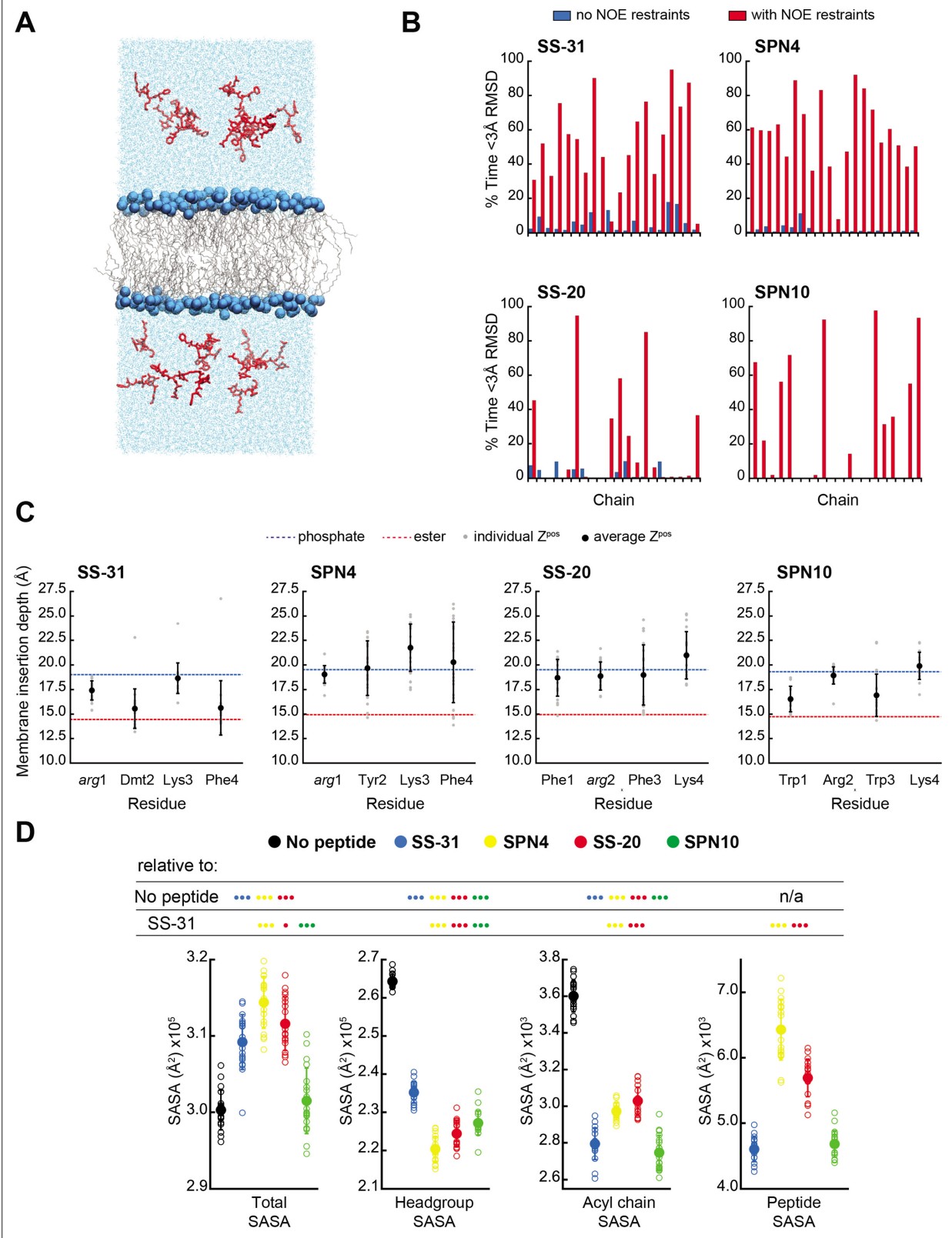

**Figure 4.** MD simulations of peptide conformations and membrane interactions. (**A**) Snapshot of a typical MD simulation. Peptides (red) are shown in the aqueous phase on either side of a bilayer composed of an 80:20 molar ratio of phosphatidylcholine (POPC):tetraoleoyl cardiolipin (TOCL) lipid acyl chains in wireframe, and lipid phosphates shown as blue van der Waals spheres. (**B**) Comparison of NMR and MD structures. Comparison of the simulation time spent by each peptide analog below a heavy atom root mean square deviation (RMSD) of 3 Å to their respective lead NMR membrane-

*Figure 4 continued on next page*

*Figure 4 continued*

bound structures before (blue) and after (red) nuclear Overhauser effect (NOE) restraints were imposed. (**C**) Average membrane insertion depths. Bilayer depth ($Z^{pos}$, n=20) for $C^\beta$ atoms on each residue for each peptide shown in comparison with the average $Z^{pos}$ levels of lipid headgroup phosphates (dashed blue) and lipid ester carbons (dashed red). The positions of the lipid atoms were averaged between the four different peptide systems for consistent comparison. The depths represent normalized distances to the bilayer COM. Gray dots represent individual points and black circles with error bars represent means ± SD. (**D**) Peptide-dependent solvent accessible surface area (SASA) during MD trajectories. SASA of the total bilayer and peptides system (Total) and individual components (headgroup, acyl chain, and peptide) calculated from MD trajectories in the absence of peptide or with the peptide analogs indicated (n=19 for all measurements). Open symbols show individual block-averaged datapoints from each trajectory and solid symbols with error bars represent means ± SD for each dataset. Statistical comparisons are based on the Wilcoxon rank sum test, with differences representing a comparison to the no-peptide control, or to SS-31 as indicated (no symbol, $p>0.05$; • $p≤0.05$; •• $p≤0.01$; ••• $p≤0.001$).

The online version of this article includes the following source data and figure supplement(s) for figure 4:

**Source data 1.** MD data: frequency of sampling states with high similarity to NMR structures, membrane insertion depths, and solvent accessible surface area (SASA) values.

**Figure supplement 1.** SS-31 side chain insertion depths from MD simulations.

**Figure supplement 2.** SS-20 side chain insertion depths from MD simulations.

**Figure supplement 3.** SPN4 side chain insertion depths from MD simulations.

**Figure supplement 4.** SPN10 side chain insertion depths from MD simulations.

**Figure supplement 5.** Comparison of peptide side chain insertion depths from MD simulations.

**Figure supplement 6.** Nuclear Overhauser effect (NOE) restraint violations from peptide-bilayer MD simulations.

**Figure supplement 7.** Time courses of solvent accessible surface area (SASA) measurements from MD trajectories.

**Figure supplement 8.** Bilayer thickness and area per lipid measurements from MD simulations.

**Figure supplement 8—source data 1.** MD data: bilayer thickness and area per lipid values.

**Figure supplement 9.** Lipid radial distribution profiles from MD simulations.

**Figure supplement 10.** Blocked SE for several analyses from MD simulations.

charge-charge interactions, long-range electrostatic effects could be missed by the ~5 Å detection threshold of the NOE.

## SPN10 minimizes membrane surface area and CL self-interactions

We next used MD simulations to obtain insights into the effects of peptide binding on bilayer properties. We first considered molecular exposure at the membrane-solvent interface by calculating the solvent accessible surface area (SASA) of different groups (*Figure 4D* and *Figure 4—figure supplement 7*). We measured the total membrane and peptide SASA of a given system in addition to its component headgroup, acyl chain, and peptide SASA values in the presence and absence of peptides. In the absence of peptides, total membrane SASA can be parsed into headgroups, which constitute the majority of solvent-exposed area, and acyl chains, whose exposure can be interpreted as interfacial lipid packing defects (*Mitchell et al., 2020*; *Boyd et al., 2018*). As expected, the presence of all peptides at the bilayer interface reduced both lipid headgroup and acyl chain SASA, due to peptides 'covering' the bilayer surface. Among the four analogs, SS-20 and SPN4 were more solvent exposed, consistent with their more superficial binding near the solvent interface (*Figure 4C*). By comparison, SS-31 and SPN10 caused the greatest decrease in acyl chain SASA, consistent with their deeper burial into the non-polar core (*Figure 4C*), filling the 'voids' of solvent-exposed hydrocarbon chains. The most notable distinction between the more deeply buried SS-31 and SPN10 was that SPN10 also caused a significantly lower headgroup SASA. Hence, SPN10-containing bilayers had the lowest total SASA among all peptide analogs, and SPN10 was the only peptide to not significantly change the total SASA compared to the bilayer-only system. The observation that total SASA is unchanged when SPN10 binds the bilayer indicates that the total amount of surface that is 'created' by this bound peptide is approximately equal to the total amount of bilayer surface area that is 'buried' by it.

Second, we considered peptide-induced elastic deformations of the bilayer, namely cross-sectional area per lipid and bilayer thickness (see *Figure 4—figure supplement 8*). Our measurements of these parameters before peptide binding (i.e. peptides restrained in solution) were consistent with previous MD work from our group (*Mitchell et al., 2020*; *Boyd et al., 2018*; *Boyd et al., 2017*) and others (*Wilson et al., 2019*). Upon peptide binding, we observed an inverse trend between bilayer thickness

(*Figure 4—figure supplement 8*, panels A and B) and mean lipid area (*Figure 4—figure supplement 8*, panels C and D). SS-31 and SPN10 expanded mean lipid area (decreased bilayer thickness) to a greater extent than did SPN4 and SS-20, which is likely related to the deeper burial of SS-31 and SPN10 aromatic side chains into the membrane (*Figure 4C*). The combined observations that SPN10 both expanded membrane area (*Figure 4—figure supplement 8*) and maintained the lowest total SASA (*Figure 4D*) suggest that even though this analog causes elastic bilayer expansion, it maintains a low 'ruggedness' of exposed membrane surface (*Tüchsen et al., 2003*).

Finally, we constructed radial distribution profiles to determine peptide-lipid and lipid-lipid interactions in the lateral (x-y) plane of the membrane (*Figure 4—figure supplement 9*). As a proxy for peptide-lipid association (*Figure 4—figure supplement 9*, upper panels), we chose the $C^{\zeta}$ atom of the Arg at the first (SS-31/SPN4) or second (SS-20/SPN10) position. We observed three radial shells of POPC and TOCL phosphates around each Arg. SS-31 and SPN4 had a greater density of CL phosphates in the closest shells, whereas CL phosphates of distal radial shells were more populated for SS-20 and SPN10. This may be related to the proximity of the Arg to the cationic $NH_3$ terminus in the B-φ-B-φ peptides causing greater local density of anionic CL. The distribution profiles of Arg-PC were, by comparison, much more similar among peptides. Our analysis of lipid-lipid radial distributions (*Figure 4—figure supplement 9*, lower panels) revealed four radial densities of lipid phosphates around corresponding phosphates of CL or PC. CL phosphates in the same Z-plane do not often approach closer than ~5 Å (*Mitchell et al., 2020*), likely due to charge-charge repulsion. Notably, SS-31 caused an increased density of CL around itself at distances <6 Å, suggesting that SS-31 may draw CL phosphates out of their respective Z-plane more than other analogs, perhaps consistent with this peptide's membrane-thinning effects (*Figure 4—figure supplement 9*). By comparison, SPN10 disfavored close contact of CL, which may be related to its large aromatic Trp side chains inhibiting close approach of sterically bulky CL.

## SPN10 has a markedly greater effect on membrane electrostatics

Our previous work showed that SS-31 modulates membrane surface electrostatics as a key part of its molecular MoA (*Mitchell et al., 2020*). We therefore sought to evaluate the effects of our four peptide analogs on membrane electrostatic potentials (*Figure 5*). We first analyzed $\psi_s$, which originates from fixed charges at the membrane interface and is strongly negative for CL-rich mitochondrial membranes (*Ohshima, 2010*; *Robertson and Rottenberg, 1983*). To this end, we used the fluorescent reporter probe 8-anilinonaphthalene-1-sulfonic acid (ANS), which reversibly binds anionic membranes, with corresponding increase in quantum yield, in a manner that is promoted by $\psi_s$ attenuation (*Robertson and Rottenberg, 1983*; *Gibrat et al., 1983*). As we have shown, ANS profiles are consistent with zeta potential readouts of membrane surface charge (*Mitchell et al., 2020*). We first measured the effect of peptide analogs on the surface charge of mitochondrial membranes by titrating mitoplasts (mitochondria with disrupted outer membrane) with peptide (*Figure 5A*). All peptides caused a saturable decrease in membrane surface charge, with SPN10 causing markedly higher attenuation (*Figure 5A*, left); by comparison, within the resolution of this assay, there was no discernible difference in saturation binding among peptides (*Figure 5A*, right). These results support that the highest $\psi_s$ downregulation is caused by SPN10 in organello, but do not rule out that this could be caused by peptide interaction with mitochondrial proteins. We therefore repeated this analysis in a more reductionist system with CL-containing LUVs (*Figure 5B*). Again, SPN10 showed the greatest effect on $\psi_s$ in this lipid-only system (*Figure 5B*, left) despite all analogs having similar saturation curves (*Figure 5B*, right). Control experiments conducted in the absence of membranes confirmed that peptides alone had no effect on ANS fluorescence (*Figure 5—figure supplement 1*). Taken together, these results show that SPN10 is a markedly more potent attenuator of membrane surface charge, originating from down-tuning of lipid bilayer surface charge.

We next analyzed membrane dipole potential ($\psi_d$), which originates from the preferential arrangement of interfacial lipid and water dipoles and contributes significantly (several hundred mV) to membrane electrostatic profiles (*Wang, 2012*; *Figure 5C*). Importantly, $\psi_d$ influences the translocation of hydrophobic ions across bilayers and may affect binding interactions with peptides (*Zhan and Lazaridis, 2012*). To evaluate the effects of our peptide analogs on $\psi_d$, we used ratiometric fluorescence excitation measurements of the membrane-bound probe di-8-ANEPPS, which has been shown to report dynamic changes in $\psi_d$, of model membranes (*Clarke and Kane, 1997*; *Hollmann*

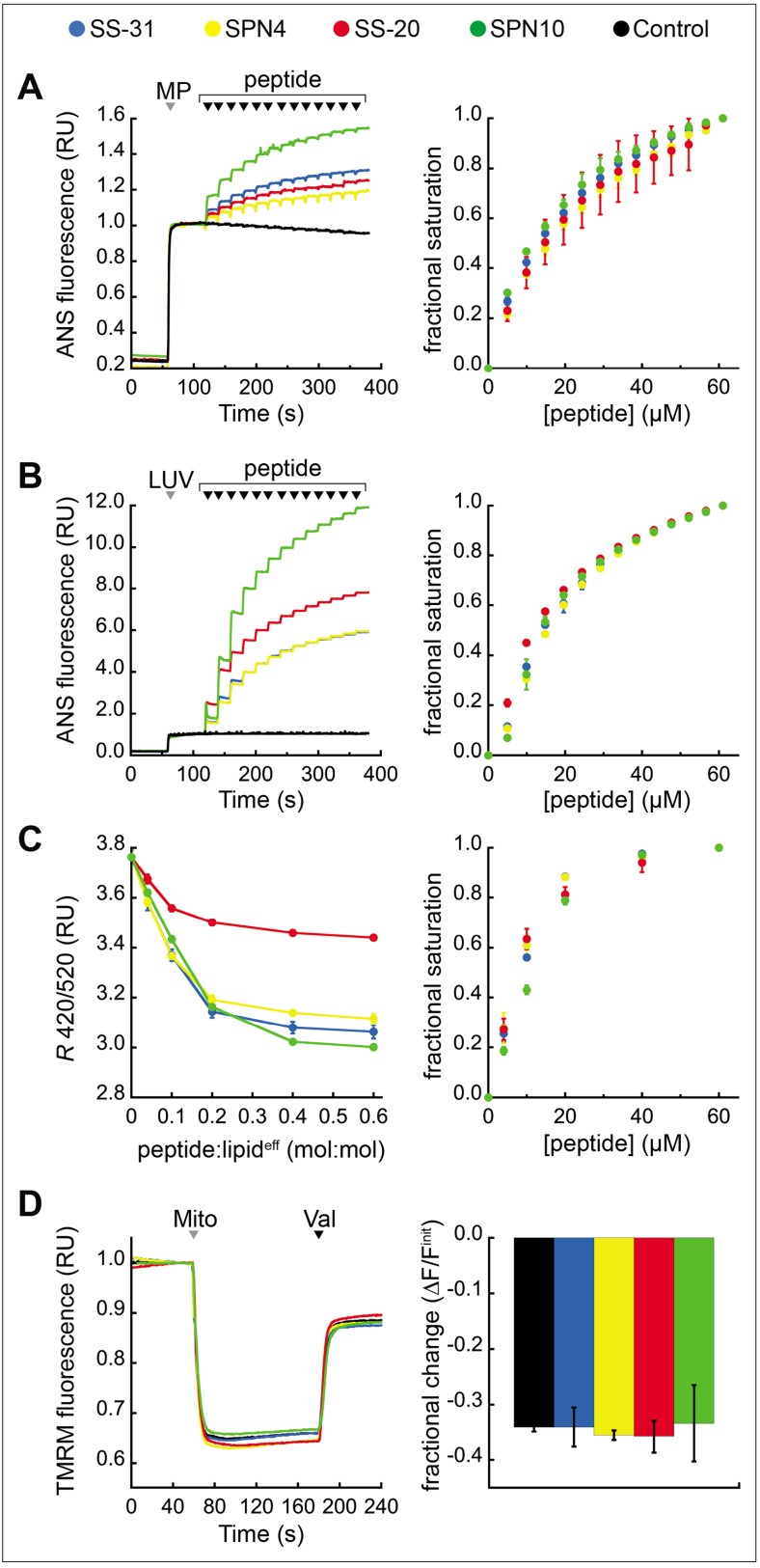

**Figure 5.** Effects of peptides on membrane electrostatic potentials. (**A,B**) Effects on surface potential ($\phi_s$). Left: signal-averaged time courses of 1,8-ANS emission with addition of (**A**) mitoplasts (generated from *Saccharomyces cerevisiae* mitochondria, 200 µg total protein) or (**B**) 80:20 phosphatidylcholine (POPC):tetraoleoyl cardiolipin (TOCL) large unilamellar vesicles (LUVs) (100 nmol lipid^eff), shown by gray arrowheads ('MP' and 'LUV', respectively),

*Figure 5 continued on next page*

*Figure 5 continued*

followed by sequential addition of peptide (10 nmol each), shown by black arrowheads. All data normalized to emission with mitoplasts or LUVs only (60–120 s). Right: saturation binding curves taken from 1,8-ANS time course data. (**C**) Effects on dipole potential ($\phi_d$). Left: profile of di-8-ANEPPS measurements (*R*, ratio of 670 nm emission from 420 nm and 520 nm excitation peaks) in the presence of 80:20 POPC:TOCL LUVs (25 nmol lipid$^{\text{eff}}$). Right: saturation binding curves taken from ratiometric di-8-ANEPPS measurements. (**D**) Effects on transmembrane potential ($\Delta\Psi_m$). Left: signal-averaged time course profiles of TMRM emission with addition of mitochondria from *S. cerevisiae* (200 µg total protein) following preincubation with 10 µM of respective peptide (gray arrowhead, 'Mito') and addition of ionophore valinomycin (black arrowhead, 'Val'). All data normalized to the pre-mitochondria addition mean (40–55 s). Right: fractional change in TMRM emission following mitochondria addition. All means and traces are from n=3 independent samples and all error bars indicate SD. Control, addition of peptide buffer vehicle only.

The online version of this article includes the following source data and figure supplement(s) for figure 5:

**Source data 1.** Membrane electrostatic parameters: ANS mitoplasts, ANS liposomes, ANEPPS, and TMRM data.

**Figure supplement 1.** Emission of 1,8-ANS with peptide titration in the absence of membranes.

**Figure supplement 1—source data 1.** ANS data with and without membranes.

*et al., 2013*). Titration of LUVs with peptides resulted in a saturable reduction in $\phi_d$ (*Figure 5C*, left) with fractional binding (*Figure 5C*, right) consistent with our ITC-based binding curves (*Figure 2*). These results indicate that all peptides cause saturable disordering of lipid and/or water dipoles upon binding. Furthermore, SS-20 had a markedly weaker effect than the others, possibly due to its comparatively lower aromatic bulk and/or lack of polar groups on aromatic side chains.

Lastly, we tested the effect of our peptides on the transmembrane potential ($\Delta\Psi_m$), which is based on ion asymmetry across the IMM established by OXPHOS proton pumping (*Mitchell and Moyle, 1967*). Both SS-31 and SS-20 have been shown to have no effect on $\Delta\Psi_m$ in healthy mitochondria (*Zhao et al., 2004*; *Mitchell et al., 2020*), and it is crucial to verify that other mitochondria-targeted peptides have no IMM-uncoupling properties. To test this, we used the potentiometric probe TMRM in quench mode, whereby probe fluorescence quenches upon accumulation in the matrix of energized mitochondria (*Perry et al., 2011*; *Malhotra et al., 2013*). No peptides had any measurable effect on the TMRM-detected magnitude of $\Delta\Psi_m$ (*Figure 5D*), indicating that they neither hyperpolarize nor depolarize the IMM under the conditions tested.

Taken together, these mitochondria-targeted tetrapeptides modulate membrane electrostatic potentials differently, in ways that depend on their side chain compositions. Most notable was the effect of SPN10 on $\phi_s$, which we propose to be a key underpinning of the activity of these peptides (*Mitchell et al., 2020*). Having addressed specific aspects of their molecular structure and behavior, we proceeded to test how these features of our tested peptides relate to their efficacy in ameliorating stress using cell culture models.

## Cellular activity of mitochondria-targeted tetrapeptides

We next tested the activity of our peptide analogs in cell culture. The pharmacological efficacy of these cationic-aromatic peptides requires that they are cell-permeable and that they target mitochondria. Although these features have been extensively demonstrated for SS-31 (B-φ-B-φ) (*Zhao et al., 2003*; *Zhao et al., 2004*) and SS-20 (φ-B-φ-B) (*Sun et al., 2020*) it is not clear if tetrapeptides with other side chain compositions would behave similarly. We therefore compared the cell uptake and mitochondrial localization of N-biotinylated SS-31 (bio-SS-31) and N-biotinylated SPN10 (bio-SPN10), in human kidney epithelial cells (HK-2) and retinal pigment epithelial cells (ARPE-19). The two biotinylated peptides readily penetrated both cell lines within an hour and showed a mitochondrial localization pattern (*Figure 6A*). These results indicate that cell uptake and mitochondrial localization are supported by either of the two possible sequence registers for these tetrapeptides.

To investigate the cytoprotective effects of our test peptides in cell culture, we used serum starvation as a stress model. Extensive studies have shown that SS-31 promotes cell survival under a variety of stress challenges, including ischemic, hypoxic, metabolic, and oxidative stress conditions, but has no effect under normal conditions (*Szeto, 2018*). Serum starvation is frequently used as a stress model in cell cultures, as it causes decreases in $\Delta\Psi_m$ and cellular ATP, cell cycle arrest, and apoptosis (*Charles et al., 2005*; *Patten et al., 2014*; *Zhu et al., 2006*; *Zhou et al., 2020*). Experiments in

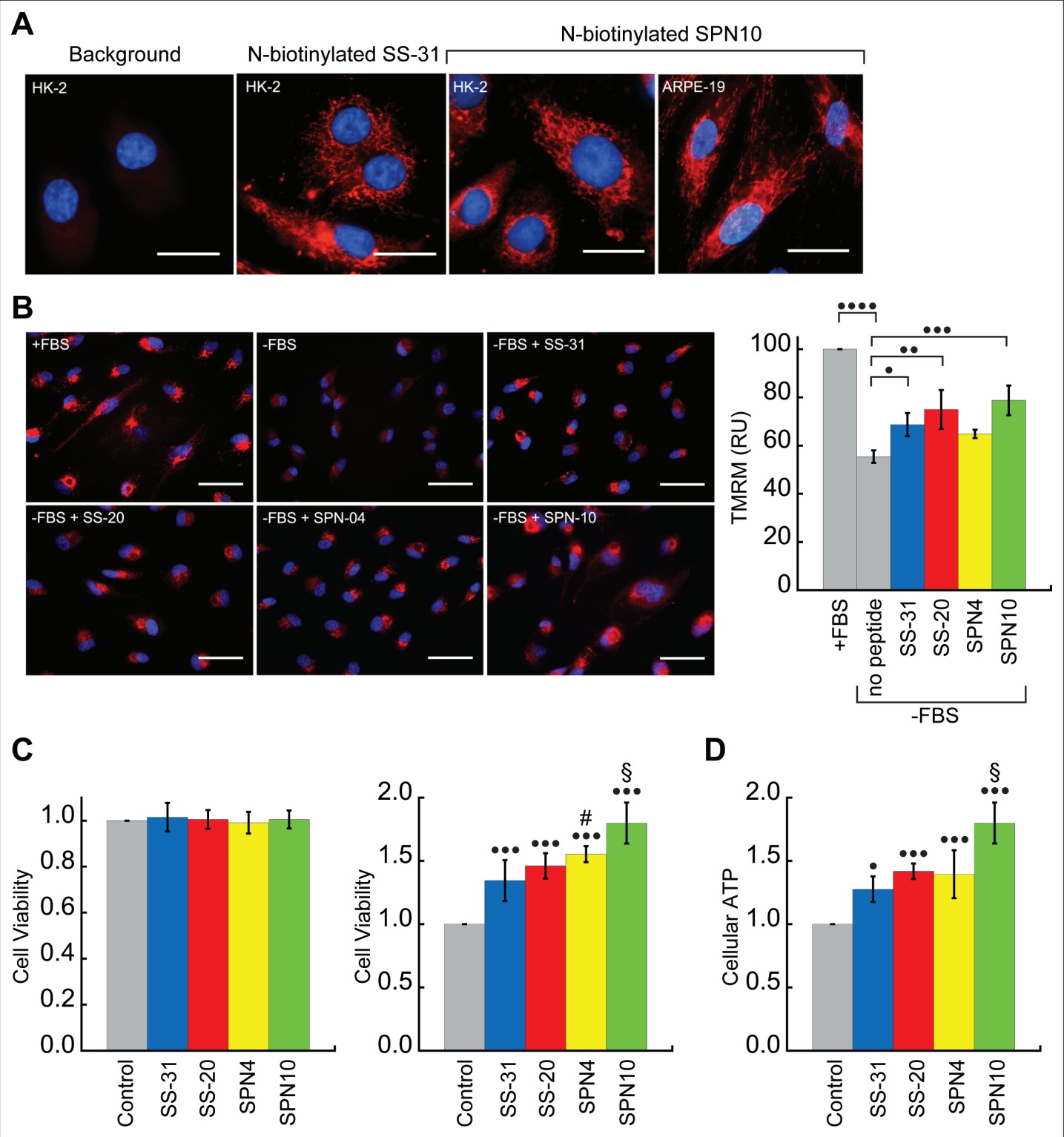

**Figure 6.** Cellular localization and efficacy of mitochondria-targeted peptides in cell culture. (**A**) Intracellular localization of peptides. Confocal microscopy images of N-biotinylated SS-31 (bio-SS-31) and N-biotinylated SPN10 (bio-SPN10) in human kidney epithelial cells (HK-2) and retinal pigment epithelial cells (ARPE-19) as indicated. Biotin was visualized with streptavidin-AlexaFluor 594 (red), and cell nuclei were stained with Hoechst 33342 (blue). Scale bars: 25 μm. (**B**) Peptide-dependent restoration of ΔΨ$_m$. ARPE-19 cells were grown with serum (+fetal bovine serum [FBS]) or subject to serum withdrawal for 3 days (−FBS) followed by incubation ± mitochondria-targeted peptides (10 nM), as indicated, for 2 hr prior to imaging. Left: representative fluorescence microscopy images show nuclei staining with Hoechst 33342 (blue) and mitochondria visualization with TMRM (red), which

*Figure 6 continued on next page*

*Figure 6 continued*

accumulates in a $\Delta\Psi_m$-dependent manner. Scale bars: 50 μm. Right: TMRM intensity was normalized to nuclear number per field. Means ± SD are from three independent experiments, and data were averaged from four to six random fields per experiment. Serum deprivation significantly reduced TMRM intensity (•••• $p\leq0.0001$) and peptide treatment restored TMRM intensity to varying levels of significance relative to the no peptide control (• $p\leq0.05$; •• $p\leq0.01$; ••• $p\leq0.001$). (**C** and **D**) Peptide-dependent protection of cell viability and ATP production. HK-2 cells grown without or with 7 days of serum deprivation in the absence or presence of mitochondria-targeted peptides (10 nM) as indicated. Reported values are normalized to the no peptide control. (**C**) Left: viability of cells grown 3 days under control conditions (Dulbecco's Modified Eagle's Medium [DMEM] supplemented with FBS); right: viability of cells subjected to 7 days of serum deprivation (DMEM only). (**D**) Cellular ATP levels of cells subjected to 7 days of serum deprivation. All means are from n=5–6 independent samples and all error bars indicate SD. Statistical comparisons show differences representing a comparison to vehicle-only control (• $p\leq0.01$; ••• $p\leq0.0001$) or comparison to SS-31 treatment (# $p\leq0.05$; § $p\leq0.001$). All statistical analyses in panels B–D represent one-way ANOVA with Tukey's multiple comparison test.

The online version of this article includes the following source data and figure supplement(s) for figure 6:

**Source data 1.** Cell culture data: TMRM, cell viability, and cellular ATP measurements.

**Figure supplement 1.** Peptide-dependent restoration of mitochondrial structure and ATP content with serum deprivation stress.

**Figure supplement 1—source data 1.** Cell culture data: cellular ATP measurements.

the present study were conducted by growing cells in serum-containing media (Dulbecco's Modified Eagle's Medium [DMEM] or DMEM/F12 with 10% fetal bovine serum [FBS]) and subjecting them to stress by growing in serum-free media (DMEM or DMEM/F12 only) for a specified length of time with or without peptide treatment.

We first tested the ability of the peptides to restore $\Delta\Psi_m$ in cells subjected to serum withdrawal. To this end, we used the potentiometric reporter TMRM in non-quench mode, whereby the probe accumulation in mitochondria is positively correlated with $\Delta\Psi_m$ (*Perry et al., 2011*; *Figure 6B*). Cells grown in the absence of serum for 3 days showed a clear mitochondrial depolarization relative to positive controls grown with serum (p<0.0001). Interestingly, three peptides showed partial restoration of $\Delta\Psi_m$ on a remarkably short timescale, within 2 hr of treatment. Following the treatment of serum-deprived cell culture with 10 nM peptide, the benchmark SS-31 caused a significant increase in TMRM signal (p<0.05). By comparison, the TMRM signal of SPN4-treated cells was not significantly higher than serum-deprived cells, whereas that of cells treated with SS-20 (p<0.01) and SPN10 (p<0.001) was significantly higher than those treated with SS-31.

We next addressed the potential restorative effects of these peptides on the structure of the mitochondrial network irrespective of their energized state. To this end, we subjected HK-2 cells to the same conditions described in *Figure 6B* and co-stained mitochondria with TMRM and MitoView Green, a probe that is widely used as a $\Delta\Psi_m$-independent marker of mitochondrial morphology (*Chaudhry et al., 2020*; *Liu et al., 2017*; *Figure 6—figure supplement 1*, panel A). Cells grown in the absence of serum showed fragmentation of the mitochondrial reticular network with perinuclear clustering in comparison with the extended network observed with non-stressed cells in the presence of serum. Consistent with the rapid kinetics observed for TMRM measurements, treatment with any of the four peptides for 2 hr before observation resulted in restoration of the reticular network pattern seen in the absence of stress. Taken together, these results strongly support that the mode of action of these peptides entails a rapid repolarization of the IMM and restoration of the mitochondrial network when challenged with serum withdrawal stress.

We then tested the ability of our peptides to promote cell viability by growing them under serum depletion conditions with and without peptide (*Figure 6C*). Control tests confirmed that when grown with serum-containing media for 3 days, the presence of peptide had no effect on viability (*Figure 6C*, left), consistent with previous observations that SS-31 has no effect under non-stressed conditions (*Szeto, 2018*). However, when cells were subject to serum deprivation conditions for 7 days, the presence of peptide strongly improved cell survival (*Figure 6C*, right); in this case, we observed marked differences in potency among the peptides, with SPN10 significantly outperforming SS-31 (p<0.001). To evaluate how our peptides affect general energy metabolism with serum withdrawal stress, we measured cellular ATP content following serum starvation for 7 days (*Figure 6D*). We observed a striking parallel with the results from our viability tests, wherein peptides show the same trends in cellular ATP with SPN10 significantly better than SS-31 (p<0.001). In addition, SPN10 significantly raised ATP levels in ARPE-19 cells with serum starvation (*Figure 6—figure supplement 1*, panel B).

Taken together, our cell culture studies show that, under the serum-free conditions tested, all four peptide analogs displayed mitochondrial protection to varying degrees. Based on our quantified cell culture measurements, the rank order of peptide analog effectiveness was SPN4 <SS-31<SS-20<SPN10 for $\Delta\Psi_m$ restoration, SS-31=SS-20<SPN4<SPN10 for cell viability, and SS-31 <SS-20=SPN4<SPN10 for cellular ATP. How these trends in stressed cells may relate to our structural and biophysical analyses will be discussed below.

## Discussion

The objective of this study was to evaluate the effects of modifying two key features of mitochondria-targeted peptides, side chain composition and sequence register, to better understand what chemical properties are important for their therapeutic efficacy. SS-31 and SS-20, which have both been validated in pre-clinical and clinical tests, feature different aromatic side chains and ordering of cationic and aromatic groups. The efficacy of these compositionally distinct peptides motivated us to test the structural and functional consequences of further modifications within the cationic-aromatic motif of these peptides. All four analogs were found to bind CL-containing bilayers, modulate membrane electrostatics, and show mitochondrial protection in cell stress models. Collectively, these observations support the concept that this general motif is sufficient for the molecular action of these mitochondria-targeted compounds. Yet there were also marked differences in their structures and molecular behaviors. These differences could provide critical insights into the MoA of this class of compounds that can be leveraged to develop more effective mitochondrial therapeutics. A summary of the key similarities and differences among the four tetrapeptides analyzed in this study is presented in *Supplementary file 6*.

A central aim of this study was the structural characterization of these peptides in solution and in the membrane-bound state. To this end, we used complementary approaches of NMR spectroscopy and MD simulations, providing the first models of the structures of these compounds. As expected, all peptides are largely disordered in solution (*Figure 1B* and *Figure 1—figure supplement 1–*) and assumed a more compact and well-defined structure when membrane-bound (*Figure 3*). This trend is reflected by the general decrease in the calculated RMSD and $R_g$ seen for the bound structures (*Figure 1C*). Intra-peptide aromatic ring stacking and cation-π interactions are stabilizing features of both the solution and membrane-bound states (*Figure 1B* and *Figure 3*). This point is significant because cation-π complexes are known to lower the energetic cost of partitioning basic side chains into the non-polar membrane core (*Hohlweg et al., 2018*; *Infield et al., 2021*), which likely contributes to the ability of these tetrapeptide analogs to traverse cell membranes, including those of the blood-brain barrier (*Zhao et al., 2003*; *Reddy et al., 2017*), and to reside stably at the membrane interface.

The most notable structural feature of these tetrapeptides is the membrane-bound reverse-turn conformation observed for all analogs other than SS-31. These conformers can all be formally classified as four-residue beta turns stabilized by an *i* to *i*+3 main chain H-bond with an Cα(1) to Cα(4) distance of ~7 Å or less (*Venkatachalam, 1968*; *Lewis et al., 1973*). The dihedral angles of the *i*+1 and *i*+2 residues of these structures deviated from the canonical phi/psi values for the most common beta turns; however, with increasing availability of high-resolution protein structures, it is becoming clear that the central residues of beta turns can occupy a wider diversity of Ramachandran space than previously appreciated (*Shapovalov et al., 2019*). The φ-B-φ-B peptides (SS-20 and SPN10) are stabilized by a CO(1) to NH(4) H-bond, whereas the B-φ-B-φ peptide (SPN4) is stabilized by an H-bond between CO(2) and the C-terminal amide group (in lieu of a 'fifth' residue) (*Figure 3*). This latter feature underscores the importance of C-terminal amidation in these compounds: not only does the amide remove the C-terminal carboxyl function to maintain the net +3 charge of these compounds, but as shown in this study, it also provides an H-bond donor to stabilize the turn structure of peptides with B-φ-B-φ register. Given that the H-bonds of these reverse turn structures reside in the low-dielectric microenvironment of the membrane interface, they are likely to be more stabilizing than when in bulk aqueous solution (*Gao et al., 2009*). By partially satisfying the H-bonding capacity of main chain atoms, this *i* to *i*+3 polar interaction partially offsets the energetic penalty of dehydration of polar backbone functional groups at the membrane, which may in turn allow the peptides to reside near the polar-apolar boundary of the membrane. From these findings, one possibility to explore is the synthesis of cyclized

forms of these peptides, given that forcing the peptides to adopt the bioactive pose may lessen the entropic penalty of membrane binding.

Our results provide several insights into the nature of the interaction between the tetrapeptides and lipid bilayers. The first insight pertains to peptide-binding density (*Figure 2—figure supplement 1*). Based on the lipid:peptide stoichiometries (*n*) from our ITC analyses (*Figure 2*), coupled with the known cross-sectional areas of POPC (70 Å$^2$, *Leftin et al., 2014*) and TOCL (129 Å$^2$, *Boyd et al., 2018*), one can calculate the binding footprint of peptides in the bilayers (80:20 POPC:TOCL) used for ITC. In increasing order, the per-peptide membrane areas are: 270 Å$^2$ (SPN10) <434 Å$^2$ (SS-31)<466 Å$^2$ (SPN4) <605 Å$^2$ (SS-20). Hence, SPN10 has a greater binding density, and SS-20 has a lower binding density compared with SS-31/SPN4. Given that the mitochondrial IMM is protein-rich with little free exposed bilayer (*Capaldi, 1982*), the greater per-peptide membrane coverage of SPN10 may relate to its enhanced efficacy by increased occupancy of the limited lipid area of mitochondria (*Figure 2—figure supplement 1*). This leads to a second related insight from this work regarding the effects of these peptides on membrane surface area. Our previous work showed that SS-31 caused a decrease in acyl chain SASA, likely related to a decrease of interfacial hydration and increased lipid packing density (*Mitchell et al., 2020*). In the present study, we found that SPN10 stood apart from the other peptides in that it minimized solvent exposure of both lipid aliphatic chains and headgroups, thereby resulting in total membrane SASA similar to membranes in the absence of peptide (*Figure 4D*). This ability of SPN10 to fill packing defects is likely related to a combination of factors, including its large aromatic side chain volume, its deeper binding in the membrane interface (*Figure 4C*), and its high surface coverage (*Figure 2*). Within the IMM, the inverted conical geometry of CL creates lateral packing defects of lamellar bilayers (*Boyd et al., 2018*), which could be related to transient pore-like defects that allow small molecule permeation (*Shinoda, 2016*) and/or accessibility of acyl groups to pro-oxidants that cause lipid peroxidation (*Bour et al., 2019*). Reducing lipid packing voids could be part of the MoA of these tetrapeptides, contributing to the observed decrease in proton leak across the IMM (*Zhang et al., 2020*) and lipid oxidative damage (*Zhao et al., 2005*; *Min et al., 2011*; *Yin et al., 2016*; *Wu et al., 2017*) that occurs with SS-31 treatment. A final related point pertains to specific peptide-lipid interactions. Our trNOE (*Figure 3—figure supplement 4*) and MD (*Figure 4C*) results both indicate that bound peptides reside in the membrane interface, likely within the boundaries of the lipid phosphate and ester groups. Yet the peptide analogs may mediate different lipid interactions. Compared with other peptides, the NOEs of SPN10 to lipid protons were notably fewer and weaker (*Figure 3—figure supplement 4*). This may be related to our MD radial distribution profiles showing that the Arg of SPN10 has a preferential lateral accumulation of POPC over TOCL (*Figure 4—figure supplement 9*). Understanding exactly how specific side chain-lipid chemical interactions relate to the efficacy of these tetrapeptide analogs will require further investigation.

Our calorimetric binding analysis showed unexpected differences in the thermodynamics of the interactions of these peptides with CL-containing membranes (*Figure 2*). All analogs had roughly equal $K_D$ (Δ*G*) values, with binding dominated by favorable changes in entropy (mean TΔ*S*/|Δ*H*| ranged from 2.7 to 6.9). However, compared with SS-31/SPN4, the binding of SPN10 and SS-20 had larger and smaller enthalpic-binding components, respectively. These features may be interpreted in terms of the origins of ΔS and ΔH for peptide-membrane interactions. First, the entropic cost of small peptide binding that comes from restricting conformational, translational, and rotational degrees of freedom (adsorption entropy) is likely to be small based on theoretical considerations (*Ben-Tal et al., 2000*). Instead, the large and favorable binding entropy we observe originates largely from the classical hydrophobic effect: membrane penetration of aromatic side chains is attendant with increased solvent mobility that accompanies the desolvation of the peptide and the release of ordered waters from non-polar acyl surfaces (*Seelig, 2004*). We propose that membrane binding of SS-20 is dominated by entropically favored burial of its aromatic Phe side chains, which lack polar groups, in the non-polar membrane core. This may facilitate deeper burial of SS-20 aromatic rings in the acyl chain region, consistent with its larger binding footprint. By comparison, binding enthalpy originates largely from polar interactions between peptide and lipid headgroups. We propose that membrane binding of SPN10 is strongly favored by polar interactions involving the indole NH hydrogen bond donors of its Trp side chains. Trp is known to remain partially in the interfacial region due to its large rigid paddle-like structure (*Aliste et al., 2003*; *de Jesus and Allen, 2013*; *de Planque et al., 1999*; *Petersen et al., 2005*; *Yau et al., 1998*), which may be consistent with its higher observed binding density.

Finally, our analysis suggests that the interaction of these peptides with membranes shows enthalpy-entropy compensation, wherein the binding energy of a congeneric series of compounds remains relatively constant due to opposing changes in $\Delta H$ and $\Delta S$ (*Fox et al., 2018*), as observed with the surface interactions of different peptides (*Wang et al., 2020*; *Meier and Seelig, 2008*). Hence, from a molecular engineering perspective, optimizing the efficacy of these tetrapeptides might be directed less toward enhancing binding affinity per se and more toward enhancing the enthalpic contribution, particularly by modulating polar contacts among aromatic groups.

This work revealed striking differences among the peptides in terms of their effects on membrane electrostatic potentials (*Figure 5*). First, the $\psi_s$ of negatively-charged membranes (biomimetic liposomes and mitoplasts) was attenuated by all peptides (*Figure 5A and B*); however, SPN10 had the greatest effect by far. This may be related in part to the higher binding density of SPN10. However, it may also be related to SPN10 uniquely having two aromatic side chains with polar (indole NH) groups that can each mediate H-bond interactions with lipid phosphates, which can alter ionization behavior and reduce headgroup charge (*Graber et al., 2018*; *Kooijman et al., 2009*). Second, the $\psi_d$ of bilayers, related to the ordering of lipid polar groups and interfacial waters, was down-regulated by all peptides (*Figure 5C*). This may be due to a general effect of bound peptide causing disorganization of interfacial water dipoles. However, as a group, those peptides with polar groups on their aromatic side chains (SPN4, SS-31, and SPN10) attenuated $\psi_d$ much more than the peptide lacking aromatic polar groups (SS-20); hence, it is possible that polar contacts with lipid mediated by aromatic side chains may alter the orientation of lipid headgroup dipoles, e.g., the P-N vector of the phosphate-choline dipole of PC (*Scherer and Seelig, 1989*). Finally, it is notable that no peptide affected the $\Delta\Psi_m$ (*Figure 5D*), supporting that they do not depolarize or hyperpolarize mitochondrial membrane potentials, consistent with the strong safety profile of SS-31 and SS-20. This point is particularly relevant in comparison with other mitochondria-targeted compounds such as the antioxidant MitoQ, which is known to cause IMM depolarization in some tissues (*Gottwald et al., 2018*; *Pokrzywinski et al., 2016*). Taken together, the effects of our peptide analogs on membrane electrostatics may be a key underpinning of their relative efficacies. Our working model posits that the tuning of $\psi_s$ is an important part of the MoA of these peptides, potentially by modulating interactions of cations and polybasic proteins with CL-containing membranes or facilitating curvature by lowering headgroup anionic repulsion (*Mitchell et al., 2020*). The fact that these peptides also affect $\psi_d$ may be relevant for their activity in mitochondria, as $\psi_d$ modifiers can modulate membrane elastic properties (*Chulkov et al., 2015*) and channel gating activity (*Pearlstein et al., 2017*). The results of the present study show that the ability of peptide analogs to modulate membrane surface charge can vary greatly with side chain composition, supporting that this could be a key functional parameter for optimization in the rational design of new tetrapeptide analogs.

Our evaluation of the four peptide analogs in cell culture studies provided the most direct insights into their potential effectiveness as mitochondrial therapeutic interventions (*Figure 6*). On one hand, the peptides showed many functional similarities in mammalian cell lines: (i) both biotinylated variants of SS-31 and SPN10 accumulated in the mitochondrial network when added to two different types of human endothelial cells (*Figure 6A*), indicating that cell permeation and mitochondria targeting are sequence-promiscuous, without strict requirement for a specific side chain register or identity of aromatic side chains; (ii) all peptides were pharmacologically active, showing some degree of protection against serum deprivation stress through a mechanism involving mitochondrial energetics and network integrity (*Figure 6B–D* and *Figure 6—figure supplement 1*); and (iii) none of the peptides affected viability of non-stressed cells (*Figure 6C*). On the other hand, the four peptides showed significant differences in our cell stress models regarding partial $\Delta\Psi_m$ restoration (*Figure 6B*), cell viability (*Figure 6C*), and cellular ATP content (*Figure 6D* and *Figure 6—figure supplement 1*).

Three points are noteworthy regarding the measured differences among the peptides in cell culture and the potential underlying mechanisms. First, peptide treatment recovered the $\Delta\Psi_m$ within a time scale of 2 hr. This suggests that the primary mode of action, at least under the conditions tested, entailed rapid repolarization of the IMM. Consistent with our findings, it has been recently shown that treatment of aged cardiomyocytes with SS-31 reduced IMM proton leak after acute 2-hr treatment (*Zhang et al., 2020*). The fast-acting effects of these mitochondria-targeted compounds do not, of course, rule out the possibility that they may also elicit longer-term changes (e.g. gene expression, protein and lipid turnover, and epigenetic modifications) following

improvements in cellular bioenergetics. Second, although the biophysical and structural work of the present study focused on lipid bilayer interactions of these peptides, SS-31 has also been shown to interact with specific mitochondrial proteins in animal models (*Chavez et al., 2020*); hence, peptide-protein interactions may contribute to the efficacy of these peptides in cell culture that would not be observed with lipid-only models. Because only the interactome of SS-31 has been analyzed to date, addressing the interactions of other tetrapeptides with mitochondrial proteins will shed light on this issue. Third, we note that the trend in which the peptide analogs restore $\Delta\Psi_m$ (*Figure 6B*) does not follow the rank order in which they improve cell viability (*Figure 6C*) and ATP content (*Figure 6D*) with serum deprivation stress. Given the numerous interconnected biochemical pathways that link $\Delta\Psi_m$ with ATP content and cell viability, it is likely that the peptides could have measurably different effects on processes not considered here that would impact mitochondrial function.

In summary, as the first structure-activity analysis for this class of compounds, this study provides new insights to guide their potential optimization. Given the complexity of membrane interactions in the molecular MoA of these compounds (*Mitchell et al., 2020*), coupled with the fact that membrane protein interactions are involved in their activity (*Chavez et al., 2020*), our limited test set of four analogs could not unequivocally address all chemical features that may enhance function. But insofar as the composition and activity of SPN10 could provide a direction for optimization, the engineering of future analogs may be guided by: (i) greater bulk of aromatic R groups (which, among proteinogenic amino acids, means emphasizing Trp); (ii) ability to form compact (reverse-turn) structures when membrane-bound; (iii) polar groups on aromatic side chains that enhance enthalpy of membrane interactions; (iv) ability to decrease SASA of lipid groups; and (v) ability to down-regulate membrane $\psi_s$. Together, modulation of these features will help pave the way for rational design of next-generation variants of this class of mitochondria-targeted therapeutic compounds.

# Materials and methods
## Reagents
Peptides SS-31, SS-20, SPN4, and SPN10 were prepared by solid-phase synthesis as trifluoroacetic acid (TFA) salts by Phoenix Pharmaceuticals (Burlingame, CA). Powder stocks were reconstituted to a concentration of 10 mM as aqueous solutions and stored at –20°C. Synthetic phospholipids were purchased as chloroform stocks form Avanti Polar Lipids (Alabaster, AL), including POPC (1-palmitoleoyl-2-oleoyl-*sn*-glycero-3-phosphocholine), DHPC (1,2-diheptanoyl-*sn*-glycero-3-phosphocholine), and TOCL (1´,3´-bis[1,2-dioleoyl-*sn*-glycero-3-phospho]-*sn*-glycerol). All lipid stocks were stored at –20°C in clear glass vials with teflon-lined cap closures. For analytical spectroscopy, fluorescent probes included 1,8-ANS, di-ANEPPS, and TMRM (Thermo Fisher, Waltham, MA). For microscopy, fluorescent probes included TMRM and MitoView Green (Biotium, Fremont, CA) and Hoechst 33342 (Novus, Centennial, CO). All solutions were prepared with ultrapure water (Millipore Advantage A10 system; resistivity 18.2 MΩ•cm @ 25 °C; total oxidizable carbon 4 ppb).

## Isothermal titration calorimetry
ITC measurements were performed based on well-established procedures (*Andrushchenko et al., 2008*) that we have previously used to measure peptide-membrane interactions (*Mitchell et al., 2020*). Solutions of peptide (titrate) and LUVs (titrant) were prepared in 20 mM HEPES-KOH, pH 7.5, and lipid-into-peptide titrations were performed with a low-volume nano-ITC microcalorimeter (TA Instruments, New Castle, DE). The calorimeter cell (volume 170 μl) contained 125–175 μM peptide, and LUVs (20 mol% TOCL/80 mol% POPC, 8 mM total lipid) were injected in aliquots of 2.5 μl (20 total injections) at time intervals of 300 s at 25°C. To account for heats of dilution, experiments were performed by the addition of titrant into solutions of buffer only, which were used for baseline subtraction. Data from dilution-corrected and integrated heat flow time courses were fit as Wiseman plots (modeled as independent, identical single binding sites), from which equilibrium binding and thermodynamic parameters ($K_d$, $n$, $\Delta H$, and $\Delta S$) were determined by non-linear regression fits (NanoAnalyze software version 3.10.0, TA Instruments).

## Fluorescence spectroscopy

Steady-state fluorescence measurements were performed with a Fluorolog 3–22 spectrofluorometer (HORIBA Jobin-Yvon, Edison, NJ) equipped with single photon-counting electronics, double-grating excitation and emission monochromators, automated Glan-Thompson polarizers, and a 450-watt Xenon short arc lamp. Measurements were made either in 4 × 4 mm quartz microcells or in 1 × 1 cm quartz cuvettes with a stir disc seated in a thermostated cell holder.

## Spectral measurements of membrane electrostatic potentials

Measurements of $\phi_s$, $\phi_d$, and $\Delta\Psi_m$ were made with LUVs (80:20 POPC:TOCL), with active mitochondria isolated from *Saccharomyces cerevisiae* as described (*Alder et al., 2008*) or with mitoplasts prepared by osmotic rupture (*Alder et al., 2008*), as indicated. $\phi_s$ measurements were performed as described using the 1,8-ANS reporter probe (*Mitchell et al., 2020*). Briefly, stirred reactions containing 0.95 µM 1,8-ANS (added from 10 mM methanol stock) and either LUVs (in 20 mM HEPES-KOH, pH 7.5 with [lipid]$^{eff}$ = 50 µM) or mitoplasts (0.1 mg/ml total protein) were titrated with stepwise additions of 10 nmol peptide over 380 s time course measurements ($\lambda_{ex}$ = 380 nm; $\lambda_{em}$ = 460 nm). $\Delta\Psi_m$ measurements were performed as described using the TMRM potentiometric probe (*Mitchell et al., 2020*). Briefly, stirred reactions of TMRM assay buffer (20 mM Tris-HCl, pH 7.5, 20 mM KCl, 3 mM MgCl₂, 4 mM KH$_2$PO$_4$, 250 mM sucrose, and 0.5% [w/v] fatty acid-free BSA), respiratory substrate (2 mM NADH), and 0.1 µM TMRM (added from 10 µM stock) were supplemented with mitochondria (0.1 µg/ml total protein) that were pre-incubated with or without 10 µM of peptide, followed by potential dissipation with 2.5 µM valinomycin over 240 s time course measurements ($\lambda_{ex}$ = 546 nm; $\lambda_{em}$ = 573 nm). $\phi_d$ measurements were performed as described using the di-8-ANEPPS reporter probe as described (*Clarke and Kane, 1997*). Briefly, LUVs were prepared by adding 1 mol% di-8-ANEPPS (from ethanol stock) to phospholipids prior to drying lipid films under nitrogen gas, hydration, and extrusion. Solutions with di-8-ANEPPS-containing LUVs (in 20 mM HEPES-KOH, pH 7.5 with [lipid]$^{eff}$ = 100 µM) were titrated with peptide at the indicated peptide:lipid molar ratios and read by excitation scans ($\lambda_{ex}$ = 380–580 nm; $\lambda_{em}$ = 573 nm; 1 nm increments and 1 s integration times), from which the ratiometric value (*R*, emission resulting 420 nm: 520 nm excitation) was used as a readout of $\phi_d$.

## MD simulations

A similar approach was used as described previously (*Mitchell et al., 2020*) where the SS-31 peptide structure was generated by modifying an extended tetrapeptide with the sequence Arg-Tyr-Lys-Phe. Coordinates were modified using the Visual Molecular Dynamics (VMD) Molefacture Plugin (*Humphrey et al., 1996*) to invert the stereochemistry of the N-terminal Arg residue from L to D and to replace the 2′ and 6′ hydrogen atoms of the Tyr side chain with methyl groups. Parameters for the 2′,6′-Dmt were modeled after the parameters of 3′,5′-dimethylphenol after running its structure through ParamChem's CGenFF server (*Vanommeslaeghe et al., 2012*). Since our initial parameterization of SS-31 (*Mitchell et al., 2020*), CHARMM36m forcefield parameters for cation-π interactions have been developed. We have included cation-π terms for SS-31 and other peptide analogs to more accurately model interactions between aromatic side chains and choline headgroups found in the bilayer interfacial region.

Tetrapeptides with amino acid sequences matching SS-20 (Phe-Arg-Phe-Lys), SPN4 (Arg-Tyr-Lys-Phe), and SPN10 (Trp-Arg-Trp-Lys) SS peptide analogs were initially generated with proper stereochemistry using the UCSF Chimera Build Structure Plugin (*Pettersen et al., 2004*). CHARMM-GUI was then used to amidate each peptide's C-terminus, obtain CHARMM36m forcefield parameters with cation-π interactions enabled for aromatic residues (*Brooks et al., 2009*; *Huang et al., 2017*; *Jo et al., 2008*; *Khan et al., 2019*; *Lee et al., 2016*), solvate each system with TIP3P water model and a 150 mM NaCl concentration, and generate simulation input files. Following the CHARMM-GUI standard protocol for solvated proteins, the peptide systems were energy-minimized using the steepest-descent algorithm for 5000 steps, followed by canonical (constant-NVT; N = number of particles, V = volume, T = temperature) ensemble equilibration for 250 ps with a 1 fs timestep. All minimization, equilibration, and production simulations were performed using the GROMACS version 2019 (*Abraham et al., 2015*; *Hess et al., 2008*). These minimized and equilibrated structures of SS-20, SPN4, and SPN10 were used in our peptide-bilayer systems.

MD simulations were then used to characterize the binding process of the four peptide analogs and to investigate their respective effects on membrane structure and dynamics. All-atom systems with explicit membrane and solvent were prepared using CHARMM-GUI with the CHARMM-36m forcefield and the TIP3P water model (**Brooks et al., 2009**; **Huang et al., 2017**; **Jo et al., 2008**; **Lee et al., 2016**; **Jo et al., 2009**). Bilayers were generated with TOCL and POPC lipids at a molar ratio of 20:80 TOCL:POPC. Each system contained a total of 150 lipids (75 per leaflet). Peptide-bilayer systems were constructed by removing solvent from the bilayer systems generated by CHARMM-GUI, resizing the box's Z-dimension to 16 nm, placing 20 peptides (10 peptides on either side) 2–3 nm away from the bilayer's headgroup region, and then re-solvating to ~75% water by mass, and raising the salt concentration to 100 mM NaCl. Each system contained ~93,000 atoms. A representation of the initial setup of the peptide-bilayer systems is shown in **Figure 4A**. Following the CHARMM-GUI standard protocol for protein-bilayer systems, all systems were energy mini-mized for 5000 steepest descent steps, followed by canonical ensemble (NVT) equilibration for 100 ps with a 1 fs timestep, 200 ps of NPT (constant-NPT; N = number of particles, P = pressure, T = temperature) equilibration with a 1 fs timestep, and ~100 ns of NPT equilibration with a 2 fs timestep. The NPT equilibration steps were performed with semi-isotropic pressure coupling and the Berendsen barostat and the Berendsen thermostat. Position and dihedral restraints were used during equilibration on the lipids and peptides to maintain lipid geometry and bilayer morphology and to prevent the peptides from interacting with the bilayers during equilibration. To enforce an equal number of peptides interacting with each side of the bilayer (10 peptides per leaflet; 7.5:1 lipid-to-peptide ratio and a 1.5:1 CL-to-peptide ratio), two inverted flat-bottom restraints in the Z-direction were placed at the bottom of the box (Z=0 nm). A restraint was placed on the peptides' Cα atoms with a force constant of 200 kJ/(mol*nm) and a radius of 3 nm, which served to maintain a constant lipid-to-peptide ratio on either side of the bilayer. A second restraint was placed on the POPC phosphates with a force constant of 50 kJ/(mol*nm) and a radius of 4.5 nm to prevent the upper and lower leaflet from drifting in the Z-direction, while still allowing for natural membrane deformations.

Production simulations were run for 2 μs and saved every 1 ps. In all production simulations electrostatic and Lennard-Jones (LJ) interactions were cut off at 1.2 nm, with electrostatics shifted from 0 nm to the cutoff, and LJ interactions shifted from 1.0 nm to the cutoff. Long-range electrostatic interactions were computed using the particle mesh Ewald method and a fourier spacing of 0.12 nm. All bilayer system production runs were simulated in the NPT ensemble using the Nose-Hoover ther-mostat, Parrinello-Rahman barostat, and semi-isotropic coupling scheme, with the temperature main-tained at 303.15 K and pressure kept at 1.0 bar with semi-isotropic coupling. The time constants for pressure and temperature coupling were 5.0 and 1.0 ps, respectively, and the compressibility value was 4.5 E-5/bar. Simulations were performed using periodic boundary conditions in all dimensions, and the simulation time step was 2 fs.

Following the 2 μs unrestrained production simulations, additional 1 μs simulations were performed using NMR-derived NOE distance-restraints. Only those NOE restraints between residues *i* to *i+2* and *i* to *i+3* NOEs were imposed in the MD simulations. By only including 'long-range' restraints, we aimed to enforce the long-distance NOEs while still allowing for natural dynamics and not over-restraining the ensemble. A force constant of 5000 kJ/(mol*nm) was applied equally to each peptide (to promote sampling of more conformations) and averaged over the ensemble of 20 peptides. Since distance restraints based on instantaneous distances can heavily reduce conformational dynamics, the 20 peptides were restrained to a time-averaged distance with a decay time of 10 ps. The initial configuration for the NMR-restrained simulations was the final configuration of the 2 μs unrestrained simulations. An example.mdp file for the 20-peptide system with NOE restraints is included in the supporting information.

Simulations of single peptides in solution were also performed. The initial peptide structures were taken from the top structure in the NMR ensemble obtained in this study. The systems were solvated with the TIP3P water model and 100 mM NaCl salt concentration. These solvated systems were energy-minimized using the steepest-descent algorithm for 50,000 steps (or a maximum force toler-ance of 1000 kJ/(mol*nm)), followed by canonical ensemble equilibration for 100 ps with 2 fs timestep, and 100 ps of NPT equilibration with a 2 fs timestep accomplished using the Parrinello-Rahman pres-sure coupling scheme and the V-rescale thermostat (**Bussi et al., 2007**). Production simulations for

each analog were run for 200 ns with a 2 fs timestep in an NPT ensemble using the Parrinello-Rahman pressure coupling scheme and the V-rescale thermostat.

## Analysis of MD simulations

The binding time dependence, membrane insertion depth, and bilayer thickness were analyzed using the MDTraj Python module (**McGibbon et al., 2015**) to process the trajectories and in-house scripts using the NumPy (**Harris et al., 2020**; **van der Walt et al., 2011**), SciPy (**Virtanen et al., 2020**), and Pandas (**Reback et al., 2021**) Python modules to perform calculations.

Average structures ($S_{avg}$) were calculated using the *gmx rmsf* function in GROMACS for each of the five conditions (**Figure 1C**) according to the following: one $S_{avg}$ from 20 lead structures (both NMR conditions), one $S_{avg}$ for each of the 20 peptides (MD: membrane, 1–2 µs time range; MD: membrane [restrained] 2–3 µs), and one $S_{avg}$ for 13 × 15 ns intervals of the trajectory (MD: solution; block averaging, see **Figure 4—figure supplement 10**; **Abraham et al., 2015**; **Hess et al., 2008**). The RMSD of the heavy atoms to their respective $S_{avg}$ was calculated for each condition's corresponding ensemble of peptide structures. The $R_g$ was calculated using the *gmx gyrate* function in GROMACS for the ensembles in each condition described above (**Abraham et al., 2015**; **Hess et al., 2008**). To estimate the limiting maximum RMSD for tetrapeptides, we simulated peptide structures in solution without any experimental distance or dihedral restraints. We obtained maximal backbone RMSDs of 1–2 Å and heavy atom RMSDs of 3–5 Å relative to their initial structures (**Figure 1—figure supplement 3**).

NOE violations were calculated using in-house Python scripts that accounted for ensemble and time averaging (**Abraham et al., 2015**; **Hess et al., 2008**; **Hess and Scheek, 2003**). The instantaneous ensemble averaged distance ($r*(t)$) was computed according to equation 1.

$$r^* \left( t \right) = \left[ \frac{1}{20} \sum_{i=1}^{20} r_i \left( t \right)^{-6} \right]^{-1/6} \quad \text{eq.1}$$

Time averaging was then performed according to equation 2.

$$\overline{r^* \left( t \right)} = \left[ \frac{1}{\tau/\Delta t} \sum_{j=t}^{t+\tau/\Delta t} r^* \left( j \right)^{-3} \right]^{-1/3} \quad \text{eq.2}$$

where $\tau$ =10 ps and $\Delta t$=1 ps. Violations were determined as those time- and ensemble-averaged pair distances which exceeded the experimentally determined upper bound NOE distances plus a buffer tolerance of 0.3 A. The list of NOEs implemented in the MD simulations and the fraction of time spent in violation of the upper bound for each NOE in each analog are presented in **Figure 4—figure supplement 6**.

SASA measurements were calculated using the *gmx sasa* function in GROMACS (**Abraham et al., 2015**; **Hess et al., 2008**), which uses the double cubic lattice method (**Eisenhaber et al., 1995**). For the SASA analyses, we defined the acyl chain region of a lipid as the carbon and hydrogen atoms below the ester carbon. The area per lipid was derived from the *gmx energy* function in GROMACS (**Abraham et al., 2015**; **Hess et al., 2008**), which outputs the change in the X-Y box dimensions over time; the latter measurement was divided by the number of lipids in one leaflet (75 lipids). The RMSD of each simulated peptide analog in solution with reference to its initial structure was calculated using the *gmx rms* function in GROMACS (**Abraham et al., 2015**; **Hess et al., 2008**). Individual peptides were used as independent samples for calculating variance in the analysis of peptide-membrane MD simulations and the NMR RMSD and $R_g$ measurements in **Figure 1C**. Block averaging (**Grossfield and Zuckerman, 2009**) was used to select decorrelated time intervals (**Figure 4—figure supplement 10**) and calculate the SD for all SASA measurements (**Figure 4D**), bilayer thickness, area per lipid, and RMSD/$R_g$ from MD simulations of a single peptide in solution. All images of the systems were created using VMD (**Humphrey et al., 1996**). All figures were created using the Matplotlib Python module (**Caswell et al., 2021**).

## NMR sample preparation

For studies on the free peptides, samples were 10 mM in peptide at pH 6. For studies on the bicelle-bound peptides, peptides were at 0.75 mM concentration, and bicelles were prepared according to

a published method (*Birk et al., 2014*). The lipids DHPC, POPC, and TOCL were mixed at concentrations of 4.5, 1.5, and 0.15 µmol, respectively, followed by drying under nitrogen gas for 20 min. After vacuum desiccation overnight, the dry lipid films were resuspended in 1 ml of de-ionized water, incubated at room temperature for 30 min, and gently swirled into solution. The peptides were added to the bicelles at a molar ratio of five peptide to one cardiolipin (TOCL), and the pH was adjusted to 5.5 with potassium hydroxide.

## NMR spectroscopy

NMR experiments were performed on a 600 MHz instrument equipped with a cryogenic probe. NMR assignments for the free peptides were obtained from 2D TOCSY ($t_m$ = 70ms), ROESY ($t_m$ = 200ms), and NOESY ($t_m$ = 150ms) spectra, where $t_m$ is the mixing time. These were supplemented with $^{13}$C-HSQC and $^{15}$N-HSQC spectra obtained for the peptides at natural isotope abundance. All data were obtained at a sample temperature of 25°C. Assignments for all four peptides are given in *Supplementary file 1*. Chemical shift deviations from random coil values (*Wüthrich, 1986*) were used to infer cation-π interactions. NMR structures of the free peptides were calculated using distance restraints obtained from ROESY spectra with $t_m$ = 200ms. For transferred NOE experiments, the free and bound peptides are in fast exchange, and trNOE experiments were performed on samples that had an excess of peptide (*Birk et al., 2014*). Consequently, chemical shifts in the presence and absence of bicelles are highly similar, and NMR assignments could be readily transferred between the two types of samples. Lipid resonances were assigned from the literature (*Liebau et al., 2017*) and are summarized in *Figure 4—figure supplement 6*. The trNOE correlations for NMR structure calculations were obtained from 2D-NOESY ($t_m$ = 150 ms) experiments. The optimal $t_m$ value was chosen from the linear portion of a NOE buildup curve (*Figure 3—figure supplement 5*), to minimize spin diffusion effects (*Post, 2003*; *Jarori et al., 1994*).

## Pulse field gradient-NMR

To address whether a representative tetrapeptide is monomeric or forms oligomers in solution, diffusion-ordered spectroscopy data on the free SS-20 peptide were collected with the Varian *Doneshot* pulse sequence (*Morris and Johnson, 2002*; *Pagès et al., 2017*). The peptide sample was 10 mM at pH 6 and a temperature of 25°C, matching conditions for NMR structure determination of the free peptides. To normalize for the effects of solvent viscosity (*Wilkins et al., 1999*), the radius of hydration for the SS-20 peptide ($R_{h,SS-20}$) was calculated using a value of 3.34 Å for the $R_{h,DSS}$ of the internal DSS reference standard (*Whitehead et al., 2022*). The calculated $R_{h,SS-20}$ value of 5.3 Å indicates that the peptide is monomeric (*Figure 1—figure supplement 2*).

## NMR structure calculation and analysis

Quantification of NOESY peak intensities and NMR structure calculations was done with the programs CCPN analysis (*Vranken et al., 2005*) and Xplor-NIH (*Schwieters et al., 2003*), respectively, on the NMRbox platform (*Maciejewski et al., 2017*). The input data for NMR structure calculations were ROE (free peptide) or trNOE distance constraints (bound peptide), together with broad dihedral restraints of φ = –90±70°, $\psi$ =60±120° to maintain backbone torsional angles of the two central residues in common regions of Ramachandran space. NMR structures were calculated using simulated annealing and refinement protocols starting from 80 initial conformers with random backbone φ, $\psi$ dihedral angles. For each peptide, the 20 lowest energy structures with no violations outside the thresholds specified in *Supplementary files 2 and 3* were kept for analysis. Structure #1 in the resulting NMR structure bundles is the closest to the ensemble mean. Electrostatic surfaces for membrane-bound peptides (*Figure 3—figure supplement 2*) were calculated with the APBS program (*Jurrus et al., 2018*). Least squares structure superpositions of the NMR bundles were calculated with the FIT routine of the program MolMol (*Koradi et al., 1996*).

## Cell culture

Human kidney epithelial cells (HK-2) and ARPE-19 were obtained from American Type Culture Collection (ATCC). All cells were authenticated by short tandem repeat profiling and routinely confirmed to be negative for mycoplasma contamination using the Myco-Sniff Rapid Mycoplasma Luciferase

Detection Kit (MP Biomedicals). HK-2 cells were cultured in DMEM and ARPE-19 cells in DMEM/F12 medium containing 1 g/l glucose and 10% FBS, 100 units/ml penicillin, and 100 µg/ml streptomycin.

### Cell uptake and mitochondrial localization of CL-binding peptides

Cell uptake of SS-31 and SPN10 was determined using N-biotinylated SS-31 and SPN10 and detected by streptavidin binding. After 3 days serum deprivation to deplete endogenous biotin, cells were treated with 1 µM biotinylated peptides for 1 hr before they were fixed with 4% PFA and incubated with Streptavidin-AlexaFluor 594 antibody (Jackson ImmunoResearch, West Grove, PA) and Hoechst 33342 (Novus Biologicals, Centennial, CO). Images were obtained with a Nikon Eclipse Ti2 fluorescent microscope using a 100× objective.

### Effects of peptides on mitochondrial potential in serum-free media

ARPE-19 cells ($5 \times 10^4$ cells) were seeded in 35 mm glass dishes in DMEM/F12 medium with 10% FBS for 24 hr. Cells were then cultured in DMEM/F12 without FBS for 3 days. Peptides (10 nM) were then added in serum-free DMEM/F12 for 2 hr. Cells were then incubated with 5 nM TMRM, 100 nM MitoView Green, and 10 µg/ml Hoechst 33342 for 15 min at 37°C. Fluorescent images were immediately obtained with a Nikon Eclipse 50i fluorescence microscope (60× objective) using FITC (MitoView Green), Texas Red (TMRM), and DAPI (Hoechst) filters. Images were collected from n=3 independent experiments, using 4–6 random fields for each treatment group, and fluorescence intensity was analyzed using ImageJ (NIH). TMRM fluorescence was normalized to Hoechst fluorescence to account for differences in cell number in each field. Differences among groups were compared by one-way ANOVA. Post hoc analyses were carried out using Tukey's multiple comparisons test.

### Effects of peptides on cell viability and intracellular ATP after 7 days of serum deprivation

HK-2 and ARPE-19 cells ($5 \times 10^3$ cells) were seeded in 96-well culture plates in DMEM (HK-2) or DMEM/F12 (ARPE-19) medium containing 10% FBS for 24 hr. Cells were then switched to corresponding serum-free medium with or without 10 nM peptides for 7 days. Medium was replaced once every 3 days. All treatments were carried out with n=4–6 independent samples for each experiment. Cell viability was measured by resazurin fluorescence using the alamarBlue assay (ThermoFisher, Waltham, MA). Cellular ATP was measured using the ApoSENSOR ATP Bioluminescence Assay Kit (BioVision, Milpitas, CA). Luminescence was measured using a microplate reader (SpecraMax iD3, Molecular Devices, San Jose, CA). Results in each experiment were normalized to serum-free control, and all data are presented as mean ± SEM from four experiments. Differences among groups were compared by one-way ANOVA. Post hoc analyses were carried out using Tukey's multiple comparisons test.

### Statistical analyses and scientific rigor

All means reported represent a minimum of n=3 independently prepared sample replicates and are reported as means ± SD or SEM, as appropriate. Statistical analyses were performed using one-way ANOVAs or Wilcoxon rank sum tests as indicated. Differences among sample populations were considered significant at $p < 0.05$.

## Acknowledgements

This work was supported by the National Institutes of Health (Grant R01-AG065879 to N.N.A. and Grant R35-GM119762 to E.R.M.), by The Barth Syndrome Foundation (2020 Idea Grant to N.N.A.) and by a charitable contribution from the Social Profit Network (to N.N.A.)

## Additional information

#### Competing interests

Hazel H Szeto: HHS does not hold any position in Stealth Biotherapeutics but has financial interests in the company. The patent related to SPN4 and SPN10 is pending and was assigned to Social Profit

Network, a public charity. HHS receives research support from Social Profit Network but has no financial interests. The other authors declare that no competing interests exist.

## Funding

| Funder | Grant reference number | Author |
|---|---|---|
| National Institute on Aging | R01-AG065879 | Nathan N Alder |
| National Institute of General Medical Sciences | R35-GM119762 | Eric R May |
| Barth Syndrome Foundation | Idea Grant | Nathan N Alder |

The funders had no role in study design, data collection and interpretation, or the decision to submit the work for publication.

## Author contributions

Wayne Mitchell, Jeffrey D Tamucci, Emery L Ng, Conceptualization, Data curation, Formal analysis, Investigation, Methodology, Validation, Visualization, Writing – original draft, Writing – review and editing; Shaoyi Liu, Data curation, Formal analysis, Investigation, Validation, Visualization, Writing – review and editing; Alexander V Birk, Data curation, Formal analysis, Investigation, Methodology, Validation, Writing – review and editing; Hazel H Szeto, Andrei T Alexandrescu, Conceptualization, Data curation, Formal analysis, Investigation, Methodology, Project administration, Resources, Supervision, Validation, Visualization, Writing – original draft, Writing – review and editing; Eric R May, Nathan N Alder, Conceptualization, Data curation, Formal analysis, Funding acquisition, Investigation, Methodology, Project administration, Resources, Supervision, Validation, Visualization, Writing – original draft, Writing – review and editing

## Author ORCIDs

Wayne Mitchell http://orcid.org/0000-0003-0871-2080
Jeffrey D Tamucci http://orcid.org/0000-0002-6194-0579
Emery L Ng http://orcid.org/0000-0002-9825-5372
Eric R May http://orcid.org/0000-0001-8826-1990
Andrei T Alexandrescu http://orcid.org/0000-0002-8425-9276
Nathan N Alder http://orcid.org/0000-0003-4474-7803

## Decision letter and Author response

Decision letter https://doi.org/10.7554/eLife.75531.sa1
Author response https://doi.org/10.7554/eLife.75531.sa2

---

# Additional files

## Supplementary files

• Supplementary file 1. NMR chemical shift assignments for the peptides evaluated[a,b].

• Supplementary file 2. Statistics for the 20 lowest-energy NMR structures of peptides in their free states.

• Supplementary file 3. Statistics for the 20 lowest-energy NMR structures of peptides in their bicelle-bound state.

• Supplementary file 4. Root mean square deviation (RMSD) and radius of gyration ($R_g$) values for NMR measurements and MD simulations.

• Supplementary file 5. Improved structural agreement between MD and NMR when imposing nuclear Overhauser effect (NOE) restraints.

• Supplementary file 6. Comparison of tetrapeptide analogs evaluated in this study.

• Transparent reporting form

## Data availability

Source data are provided as follows: Main text, Figure 1: RMSD vs $R_g$ scatter plot data of Fig 1C are compiled as Excel files and uploaded as Source Data linked to Fig. 1. Main text, Figure 2: Data from

binding curves (Q [raw data, means and SD], Fig. 2A) and summary of equilibrium binding parameters (raw data, means and SD, Fig. 2B) are compiled as Excel files and uploaded as Source Data linked to Fig. 2. Main text, Figure 4: Data from chain-specific %Time <3Å RMSD +/- NOE restraints (Fig. 4B), membrane insertion depths (raw data, means and SD, Fig. 4C), and SASA analysis (raw data, means and SD, Fig. 4D) are compiled as Excel files and uploaded as Source Data linked to Fig. 4. Main text, Figure 5: Data from ANS analyses (time courses and fractional saturation [raw data, means and SD] for model membranes and mitoplasts, Fig. 5A,B), ANEPPS analyses (ratiometric values [raw data, means and SD] and fractional saturation [raw data, means and SD], Fig. 5C) and TMRM analysis (time courses and fractional change [raw data, means and SD], Fig. 5D) are compiled as Excel files and uploaded as Source Data linked to Fig. 5. Main text, Figure 6: Data from TMRM quantitation (raw data, means and SD, Fig. 6B), cell viability (raw data, means and SD, Fig. 6C) and cellular ATP (raw data, means and SD, Fig. 6D) are compiled as Excel files and uploaded as Source Data linked to Fig. 6. Additional data files: Molecular dynamics input and data files are available at: https://github.com/MayLab-UConn/SSpeptides_qsar_MDdata, (copy archived at swh:1:rev:4dec852d63fb9acc096f11573f9d1a5a7083aa48).

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
