## [Editor Report]

A number of S-S (Szeto-Schiller) tetrapeptides are known to be targeted to mitochondria. This study shows for the first time a structure-activity relationship for these peptides in reversing mitochondrial membrane potential and ATP loss in stressed cell models. In particular, peptides containing indole residue in the side chain, such as Tryptophan are more active. These results therefore provide valuable new insight on the development of new therapeutics for the treatment of mitochondrial dysfunctional diseases.

---

## [Decision Letter]

**Decision letter after peer review:**

Thank you for submitting your article "Structure-Activity Relationships in the Design of Mitochondria-Targeted Peptide Therapeutics" for consideration by *eLife*. Your article has been reviewed by 2 peer reviewers, and the evaluation has been overseen by a Reviewing Editor and Mone Zaidi as the Senior Editor. The following individual involved in review of your submission has agreed to reveal their identity: Naresh Babu V. Sepuri (Reviewer #2).

The reviewers are of unanimous opinion that the structural part of tripeptides targeted mitochondria and their association with membrane is well done. However, results on the activity part of the MS are nearly non-existant. We feel that additional data on activity relationship is required before the paper can be considered for publication in *eLife*.

The reviewers acknowledge the robustness of the data on the structures of various tripeptides and the nature of side chains on targeting efficiency and membrane association. It is also interesting that many of these peptides induce mitochondrial ATP pools. However, a crucial point on mechanism of increased ATP pool is missing. Is it due to direct binding of targeted peptides to mitochondrial electron transfer complexes? If so, is mitochondrial respiratory control affected? The authors should provide data on these important questions. These questions are particularly important for assessing the clinical relevance of the MS.

The manuscript also fails to mention about a major pathway of targeting inhibitors, antioxidants and drugs to mitochondria that has been used by many groups for targeting compounds and clinically important drugs to mitochondria. It is important to site Murphy's and Kalyanaraman's work and discuss how the tripeptide system may be superior to the TPP+ targeting system.

We collectively feel that inclusion of additional data on above mentioned aspects would significantly strengthen the manuscript and therefore return the MS for updating.

*Reviewer #1 (Recommendations for the authors):*

i. Description of NMR structural analyses is rather specialized (including the use of acronyms), and difficult to follow by the readers without appropriate expertise.

ii. Figure 1C – RMSD value for SS-20 seems similar to those of SS-21 and SPN4. Therefore, the order of amino acids does not seem to be the major predicting feature for RMSD (structural precision/dynamics) in solution. In addition, in membranes all four peptides show low values of RMSD.

iii. The experiments testing the effect of the peptides on membrane electrostatic potentials would benefit from appropriate controls to demonstrate that the increase in AND fluorescence in fact is due to changes in membrane potential, e.g. showing the effect of the peptides on ANS fluorescence in the absence of membranes (mitoplasts, LUVs).

iv. In the cell protection assays, the authors used cells "starved" of serum as a stress model. There is no clear justification for this model and how relevant it is as a predictive tool for in vivo efficacy of the compounds when used to treat mitochondrial diseases.

v. Alamar Blue viability assay – what is being measured is resorufin fluorescence rather than resazurin fluorescence.

vi. Inclusion of additional cell health and cell death assays may strengthen the conclusions.

*Reviewer #2 (Recommendations for the authors):*

It is crucial to address how the tested peptides improve mitochondrial function. SS-31, which carries unusual amino acid, quenches the ROS and might be helping in the restoration of mitochondrial health. These peptides lack this amino acid, and it would be interesting to check cellular ROS levels in the presence and absence of these peptides. How do these peptides increase cell survival and ATP production without quenching ROS? Besides, some of the following points shall be addressed to improve the manuscript.

1. As these peptides also have hydrophobic residues, are they prone to form aggregates in higher concentrations?

2. In Fig: 6A, Mitotracker or mitochondrial marker will be helpful to show that peptides are all localized to mitochondria.

3. Since there is a significant change in cellular ATP levels (in Figure 6C & 6D), authors may consider doing Oxygen consumption rate to rule out the non-mitochondrial ATP (Glycolysis).

---

## [Author Response]

Reviewer #1 (Recommendations for the authors):i. Description of NMR structural analyses is rather specialized (including the use of acronyms), and difficult to follow by the readers without appropriate expertise.

We removed some of the acronyms by shortening the NMR section of the main text. Where NMR acronyms were necessary, in the revised manuscript we fully spelled out the term in the first instance it is used.

ii. Figure 1C – RMSD value for SS-20 seems similar to those of SS-21 and SPN4. Therefore, the order of amino acids does not seem to be the major predicting feature for RMSD (structural precision/dynamics) in solution. In addition, in membranes all four peptides show low values of RMSD.

The conclusion from the NMR-based work that peptides with a B- φ-B- φ motif may have some observed residual structure in solution compared with φ-B- φ-B peptides was based on the combined observations that the former had both lower measured RMSD values *and* non-sequential NOEs. The reviewer is correct that SS-20 does not follow this observation with respect to RMSDs. We have therefore clarified this point in the Results section.

To the second point, we believe that any effect of sequence register on residual conformation will be more apparent in solution. When the peptides are bound to membranes, the observed conformers are dictated by electrostatic and nonpolar interactions with the bilayer.

iii. The experiments testing the effect of the peptides on membrane electrostatic potentials would benefit from appropriate controls to demonstrate that the increase in AND fluorescence in fact is due to changes in membrane potential, e.g. showing the effect of the peptides on ANS fluorescence in the absence of membranes (mitoplasts, LUVs).

This is an excellent point. In our previous work (Mitchell, et al. [2020] *J. Biol. Chem.* 295: 7452-7469), we confirmed that SS-31 by itself did not affect ANS fluorescence. We have performed these controls for all four peptides analyzed in this study (Figure 5 —figure supplement 1 in the resubmitted manuscript), confirming that none of the peptide analogs causes a change in ANS emission yield in the absence of membranes.

iv. In the cell protection assays, the authors used cells "starved" of serum as a stress model. There is no clear justification for this model and how relevant it is as a predictive tool for in vivo efficacy of the compounds when used to treat mitochondrial diseases.

On this point, we must respectfully disagree. There are in fact numerous studies using serum deprivation as a means to induce apoptosis and mitochondrial signaling pathways (Charles, et al. [2005] *Glaucoma* 46: 1330-1338; Bialik, et al. (1999) *Circ. Res.* 85: 403-414; Wang, et al. [2017] *J. Transl. Med.* 15, 33.), oxidative stress (White, et al. [2020] *Sci. Rep.* 10, 12505) and alteration of mitochondrial morphology (Zhou, et al. [2020] *Autophagy* 16: 562-574).

v. Alamar Blue viability assay – what is being measured is resorufin fluorescence rather than resazurin fluorescence.

We thank the reviewer for noticing this error and it has been corrected.

vi. Inclusion of additional cell health and cell death assays may strengthen the conclusions.

Our additional cell culture-based assays (see response to Point #1), specifically the new assays evaluating mitochondrial membrane potential, strengthen our conclusions about the relative potency of the peptide analogs.

Reviewer #2 (Recommendations for the authors):It is crucial to address how the tested peptides improve mitochondrial function. SS-31, which carries unusual amino acid, quenches the ROS and might be helping in the restoration of mitochondrial health. These peptides lack this amino acid, and it would be interesting to check cellular ROS levels in the presence and absence of these peptides. How do these peptides increase cell survival and ATP production without quenching ROS? Besides, some of the following points shall be addressed to improve the manuscript.

There are many past studies demonstrating that peptides lacking the 2,4-dimethyltyrosine moiety, such as SS-20, can reduce ROS burden originating from mitochondrial OXPHOS machinery (e.g., Birk et al. [2015] *BBA Bioenerg.* 1847: 1075-1084; Szeto et al. [2015] *Am. J. Physiol. Renal Physiol.* 308: F11-F21; Sun et al. [2020] *Aging* 12: 1823818250). It should be noted that a therapeutic compound can act as an antioxidant not only through scavenging ROS, but also by keeping ROS from forming in the first place (e.g., by enhancing efficiency of electron transport complexes, by promoting supercomplex assembly and stability, and/or stabilizing cristae morphology). So these peptides could in principle be curtailing ROS either before or after such compounds are produced. Importantly, in our starvation stress response studies, the peptides may be functioning in a way that does not involve ROS curtailment at all, such as mitigating proton leak across the inner membrane. This is discussed in the revised version of the paper.

1. As these peptides also have hydrophobic residues, are they prone to form aggregates in higher concentrations?

In response to the reviewer’s point, we addressed this question experimentally using NMR translational diffusion. The size of the SS-20 peptide from the diffusion measurements (Figure 1 —figure supplement 2 of the revised manuscript) is consistent with it being a monomer under the conditions of the structural studies for the free peptide (10 mM peptide concentration, pH 6, 25^o^C). We expect from our results with SS-20 that the other three peptides are monomers in solution because all four of our peptides have reasonably similar hydrophobic character, judged by several calculated qSAR parameters, including LogP(o/w), the log of the octanol-water partition coefficient; qPolariz, the predicted polarizability; and FOSA, the hydrophobic component of the total solvent accessible surface area (attached H and saturated C) (calculated by Schrödinger Maestro QikProp calculations):

**Author response table 1. sa2table1:** 

Peptide	LogP(o/w)	qPolariz (Å^3^)	FOSA (Å2)
SS-31	-1.868	55.768	326.69
SS-20	-1.852	54.319	278.69
SPN04	-2.278	55.375	234.03
SPN10	-1.798	57.984	243.73

2. In Fig: 6A, Mitotracker or mitochondrial marker will be helpful to show that peptides are all localized to mitochondria.

The colocalization of compounds belonging to this peptide class has been shown previously (e.g., in Delco et al. [2018] *J. Orthopaedic Res.* 8: 2147-2156) using MitoTracker and bio-SS-31 (the same variant used in the present study) and visualizing with streptavidin staining by confocal microscopy. Given that: (i) there is precedent for such colocalization studies with biotinylated peptide analogs and mitochondria-staining dyes, and (ii) the biotinylated variants in the present study (bio-SS-31 and bio-SPN10) show patterns consistent with mitochondrial localization, we feel that there is ample evidence to support mitochondrial localization of this class of tetrapeptides.

3. Since there is a significant change in cellular ATP levels (in Figure 6C & 6D), authors may consider doing Oxygen consumption rate to rule out the non-mitochondrial ATP (Glycolysis).

The revised manuscript includes measurements of mitochondrial membrane potential to evaluate the ability of peptides to restore mitochondrial energetics following serum deprivation stress (Figure 6B). These measurements were taken using an established fluorescence microscopy-based approach with the potentiometric probe TMRM. Our new results confirm that the peptides not only partially restore the significant membrane depolarization following starvation stress, but do so with exceptionally fast kinetics (on a ~2 h timescale) that to our knowledge has not been observed with any other mitochondria-targeted therapeutic.

Membrane potential is arguably the single most critical determinant of proper mitochondrial function because it encompasses both respiratory chain activity and membrane coupling efficiency. We therefore believe that membrane potential measurements represent a more direct way to assess mitochondrial health than respirometry measurements of oxygen consumption.

The fact that these peptides show partial recovery of membrane potential strongly supports a model in which they work through a mitochondrial mode of action. Furthermore, the fast kinetics of membrane potential restoration sheds new light on the mechanism of these peptides, at least with respect to serum deprivation stress, that likely includes rapid restoration of membrane coupling (e.g., by decreasing transmembrane proton leak). These ideas will be pursued by our team in subsequent studies.

Importantly, these new experiments are based on a single time point (two hours post peptide addition), and with a single peptide concentration (10 nM). Hence, while these results are sufficient to address the reviewer’s question regarding a mitochondria-related mode of action, they do not represent a comprehensive kinetic analysis of the pharmacological activity of these peptide analogs. A more detailed assessment of this issue will be forthcoming in future studies.